

# Assimilation of surface pressure observations from personal weather stations in AROME-France

Alan Demortier[a], Marc Mandement[a], Vivien Pourret[a], and Olivier Caumont[a,b]

[a]CNRM, Université de Toulouse, Météo-France, CNRS, Toulouse, France
[b]Météo-France, Direction des opérations pour la prévision, Toulouse, France

**Correspondence:** Alan Demortier (alan.demortier@meteo.fr)

**Abstract.** Spatially dense surface pressure observations from personal weather stations (PWSs) have proved their ability to describe physical meteorological patterns, such as convective events, which are not visible with the only use of standard weather stations (SWSs). In this study, the benefit of assimilating PWS observations with the 3DVar and the 3DEnVar data assimilation schemes of the AROME-France model is evaluated over a 1-month period and during a heavy precipitation event in the south of France. Observations of surface pressure from PWSs are bias-corrected, quality-controlled, and thinned with a spacing equal to the horizontal dimension of an AROME-France grid cell. Over France, almost half of the 55 187 available PWS observations are assimilated, which is 129 times more than the number of assimilated SWSs observations. Despite the small advantages found from their assimilation with the 3DVar assimilation scheme, the 3DEnVar assimilation scheme shows systematic improvement and reduces by $-10.3\%$ the root-mean-square departure in surface pressure between 1 h model forecasts and SWS observations over France. Significant improvement is observed up to 9 h of forecast in mean sea level pressure. Finally, when PWS observations are assimilated with the 3DEnVar assimilation scheme, a surface pressure anomaly generated by a mesoscale convective system – observed by PWSs and not visible without them – is successfully assimilated. In that case, the forecasts of location and temporal evolution of the mesoscale convective system as well as rainfall are closer to the observations when PWS observations are assimilated.

## 1 Introduction

Amongst all the challenges to improve data assimilation (DA) for convection-permitting numerical weather prediction (NWP), the use of new observation types has emerged (Hu et al., 2022). Indeed, accurately modelling the initial state (analysis) of a convective-scale model requires the assimilation of spatially dense and temporally frequent observations (Gustafsson et al., 2018). A step forward in improving analyses could come from near-surface crowdsourced observations, e.g. from smartphones or personal weather stations (PWSs) (Hintz et al., 2019). Crowdsourced observations are much more numerous than observations of standard weather stations (SWSs) currently assimilated in NWP models, while having a more heterogeneous quality and less metadata.

Clark et al. (2018) and Mandement and Caumont (2020, hereafter MC20) showed that PWS observations, when combined to SWS observations, reveal rapid spatio-temporal variations in screen-level temperature, relative humidity, and surface pres-



sure associated with the life cycles of thunderstorms which are partly visible, or not visible at all, with SWS observations only. Amongst these three observed variables, MC20 showed that PWS surface pressure observations allow multiplying by an order of magnitude of 100 the total number of available surface pressure observations compared to the current state. These observations, when simply spatialized, are the observations that most reduce the root-mean-square error of mean sea level pressure (MSLP) analyses during stormy days, in comparison with analyses only made with SWSs. The shape and amplitude

of surface pressure perturbations, which vary from one thunderstorm to another, are described by Madaus and Hakim (2016) using idealised simulations for the particular case of isolated thunderstorms. The interest in assimilating PWSs observations into a convection-permitting NWP model has therefore emerged.

As well as varying on a fine scale with the passage of thunderstorms, surface pressure is a crucial variable in NWP models. Surface pressure observations, provide information not only near the surface, but over the entire air mass in a column of the

atmosphere, and are less dependent on the surface characteristics (e.g., cities, vegetation; Ingleby, 2015), than other surface observations. Most global and limited-area models assimilate surface pressure observations over land and oceans from the World Meteorological Organization (WMO) Regional Basic Observing Network (WMO, 2021) and exchange them through the WMO Global Telecommunication System (WMO, 2020). In the Met Office global model, Ingleby (2015) showed that surface data contribute to 16.9 % of the total observation impact of 24 h forecast (compared to 10.0 % for radiosondes and

9.2 % for aircraft). In the Météo-France global model ARPEGE, Chambon et al. (2023) showed that surface observations are crucial to reduce forecast errors: over Europe, the Performance Index 18 (IP18) forecast score (based on the arithmetic average of the geopotential, temperature, and wind root-mean-square errors against radiosoundings at 48 and 72 h range and normalized by the value of the year 2008) diminished from 6.51 to −1.63 when the surface observations are removed over a 6-month period (Chambon et al., 2023), i.e. being worse than 2008 score. Surface pressure errors are identified, amongst the

other surface observations, as the main source of 24 h forecast errors in terms of a moist energy norm.

These last years, several studies (Madaus et al., 2014; McNicholas and Mass, 2018b; Hintz et al., 2019; Ridal et al., 2019) reported assimilation experiments of crowdsourced surface pressure observations from dense networks to capture local-scale features. Various techniques were designed depending on the kind of observations (e.g., smartphones or PWSs) and the assimilation system.

Among these studies, Madaus et al. (2014) assimilated between 1300 and 1700 pressure observations from PWSs over the Pacific Northwest in addition to the 140 SWSs commonly assimilated in NWP models. One-month assimilation experiments were done in the WRF model using an ensemble Kalman filter DA system. With appropriate bias correction and quality control, they showed significant improvements of 3 h forecasts of surface pressure, and lower but still significant improvements of 2 m-temperature and 10 m-wind meridian component. Furthermore, results showed a better timing of frontal passages (20

to 45 min of improvement) in the forecasts. Then, a few experiments have been done using smartphone observations. Hintz et al. (2019) showed the difficulties of the task due to quality control and bias correction issues because the altitude of the smartphone pressure observations were not accurately known (accentuated by the moving location of the sensors) and have used a 3DVar assimilation system. However, McNicholas and Mass (2018a) developed a machine learning quality control method for smartphone pressure observations. Once applied, quality-controlled observations were assimilated in an ensemble DA



system during two specific cases (McNicholas and Mass, 2018b). Results showed an improvement of the 1 h forecast on both
       2 m temperature and humidity; and on the track and intensity of a coastal wind storm. In the same way for PWSs, Ridal et al.
       (2019), implemented and compared different quality control methods, using in particular a variational bias correction to remove
       the pressure bias from the observation time series. The results of the three-week assimilation experiments in HARMONIE-
       AROME, the limited-area NWP model developed within the ALADIN consortium (Bengtsson et al., 2017), which uses a
3DVar DA scheme, did not show any advantage. In summary, the assimilation of dense surface pressure observations from
       a few pairs of observation network – DA scheme has been tested, but there is a lack of information on the role that the DA
       scheme plays in making the most of dense PWS surface pressure observations.

       Different DA schemes have been developed for AROME-France. AROME-France is the convection-permitting, limited-
       area model operational at Météo-France since 2008. AROME-France has a three-dimensional variational (3DVar) DA scheme
launched hourly and assimilates, in addition to observations assimilated in ARPEGE, dedicated observations for the mesoscale
       within a 1 h assimilation window (Seity et al., 2011; Brousseau et al., 2016). The background error covariance matrix deter-
       mines how the information from the observations is spread in the model space (horizontally and vertically) to make the analysis.
       In the 3DVar scheme, the background error covariance matrix is climatological, spatially homogeneous, and isotropic. Mont-
       merle et al. (2018) and Michel and Brousseau (2021) developed an ensemble DA scheme, to replace the static matrix with a
dynamic background error covariance matrix, improving the spread of the information from the observation. This scheme is
       called three-dimensional ensemble variational (3DEnVar).

       This study intends to answer the following questions: does the assimilation of PWSs surface pressure observations in the
       current AROME-France 3DVar data assimilation system improve weather analyses and forecasts? And what are the differences
       with the 3DEnVar system currently under development?

The remainder of this paper is organized as follows. Section 2 describes the crowdsourced surface pressure observations used
       and the different assimilation configurations available for AROME-France. The observation pre-processing and the DA exper-
       iments are described in Sect. 3. Section 4 presents the scores of one-month cycled assimilation experiments, and a case study
       of a heavy precipitation event (HPE) is analysed (Sect. 5). Finally, the advantages of the 3DEnVar system for the assimilation
       of dense surface pressure observations are discussed (Sect. 6).

## 2   Data, tools, and methodology

       A study period of 1 month is chosen, ranging from 6 September 2021 00:00 UTC to 5 October 2021 23:00 UTC. In total, it
       represents 717 cycled hours of simulation for each experiment (without two time steps during which PWS observations are
       missing). This period includes several rainy situations, such as precipitation events in the south-west of France on 8 September,
       in the south-east of France on 14 September, and over the north and south-east of France between 4 and 5 October.



## 2.1 Surface pressure observations

Pressure observations are mainly retrieved at the surface with weather stations. Two main surface networks are used, the standard weather stations and the personal weather stations.

### 2.1.1 Standard weather station observations

Standard weather stations gather land surface synoptic stations (sending SYNOP weather reports through the WMO Global Telecommunication System, see World Meteorological Organization (2020)), the French extensive data acquisition and weather observation network (RADOME, Tardieu and Leroy, 2003) and stations located over sea on ships (sending SHIP reports). On average during the one-month study period, 1124 surface pressure observations per hour are available in the AROME-France domain coming from manual land SYNOP (10.0 %), automatic land SYNOP (58.6 %), French RADOME (17.9 %), manual SHIP (2.0 %) and automatic SHIP (11.4 %) reports. Among these reports, 214 are located in Metropolitan France over land: 10.1 % are automatic land SYNOP and 89.9 % are French RADOME. Stations associated to these surface pressure observations are further called SWSs. Most of the scores will focus on SWSs located in France.

### 2.1.2 Personal weather stations observations

Amongst all types of Personal Weather Station (PWS) networks, the Netatmo network stands out being at the same time the largest in France and available in near-real time (MC20). The stations are privately owned, which explains why they are not spatially equally distributed (Fig. 1) i.e., they are denser in the cities.

Netatmo PWS is composed of outdoor and indoor modules. The indoor module measures the surface pressure. Observation times are not exactly regular: the mean time step between two observations is approximately 5 min, and the company updates its servers only every 10 min. The owner of the station can perform on-demand measurements, which shift the following observation times. Also, the evolution of the surface pressure related to the passage of thunderstorms is of the order of a few minutes. For all these reasons, to limit time lags, the observations are linearly interpolated at fixed times (i.e., round hours) using the two closest measurements around the interpolation time. If the two closest measurements are separated by a period of 700 s or more, the observation is removed.

The absolute accuracy of the pressure sensor of the Netatmo PWS given by the manufacturer is ±1 hPa (Netatmo, 2023). In 2018, the sensor was an STMicroelectronics LPS25HB, which has the same absolute accuracy indicated by Netatmo and a relative accuracy of ±0.2 hPa after calibration for standard atmospheric surface conditions (STMicroelectronics, 2016). The exact model of the sensor may change regularly, however the absolute accuracy provided by Netatmo has not changed since 2019 (MC20), suggesting that their characteristics remain similar. Once the PWS is installed and running, pressure sensor calibration is possible but is not performed by default, which can lead to systematic errors. When PWS MSLP observations were compared to SWS analyses taken as reference, MC20 showed that the first (Q25) and the third (Q75) quartiles of the systematic errors were equal to −4 hPa and 2 hPa respectively. The supplementary errors result from lagged-time pressure



Natural Hazards
and Earth System
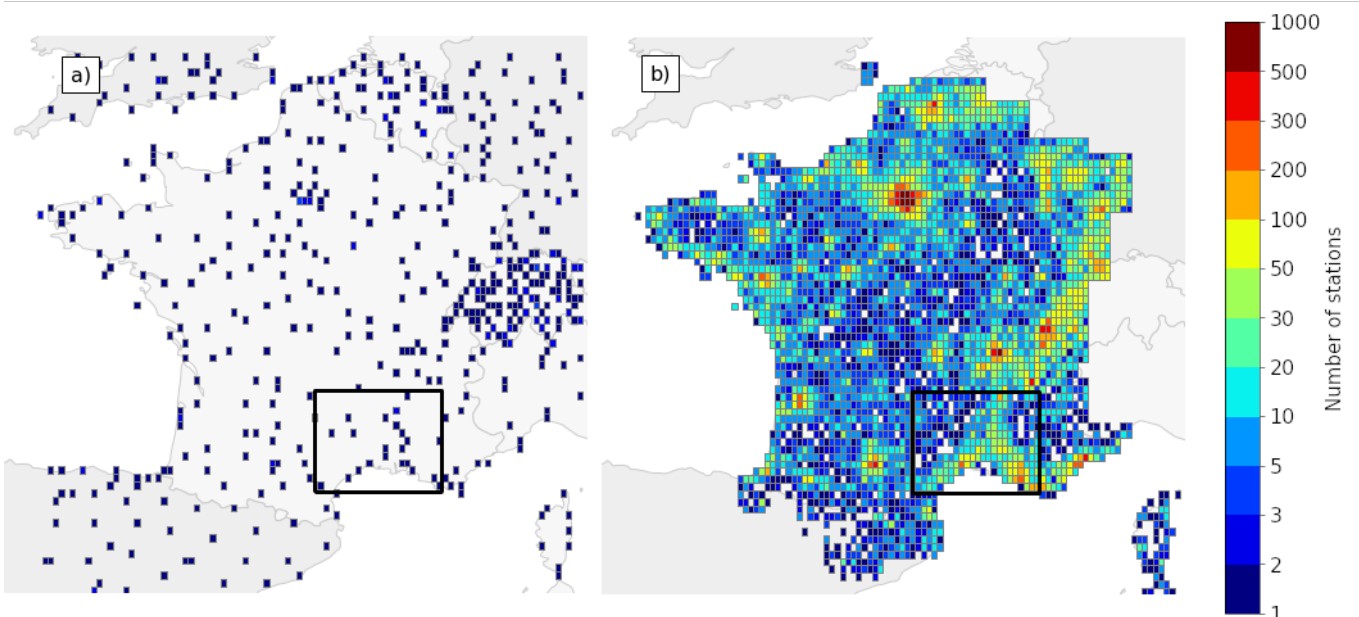

**Figure 1.** Number of (a) SWSs and (b) PWSs emitting at least one observation during the one-month study period over metropolitan France. Observation counts are binned into approximately 0.15°×0.1° bins. The black rectangle delimits the domain used for the case studied in Sect. 5.

measurements to pressure change, and inaccurate or unavailable metadata. These constraints will be taken into consideration in the pre-processing method.

## 2.2 AROME-France assimilation system

AROME-France is the Météo-France operational limited-area model (Seity et al., 2011; Brousseau et al., 2016). The model
is convection-permitting at 1.3 km horizontal grid spacing with 90 vertical hybrid pressure levels. It does not rely on the hydrostatic assumption and fully resolves elastic Euler equations. Both the physics and the surface parametrizations from SURFEX model are taken from the Meso-NH model (Lac et al., 2018). The model is regularly updated with new features, creating new development cycles. In the current study, the cycle 48t1 of AROME-France is used. Thereafter, AROME will systematically refer to AROME-France.

The DA scheme combines the active observations (see below for the meaning of active) with the background (for AROME the 1 h forecast starting from the previous analysis), to produce $x_a$ the analysis state of the atmosphere, obtained by the minimization of the cost function $J$:

$$J\left(\boldsymbol{x}\right) = \frac{1}{2}\left(\boldsymbol{x} - \boldsymbol{x}_b\right)^{\mathrm{T}}\mathbf{B}^{-1}\left(\boldsymbol{x} - \boldsymbol{x}_b\right) + \frac{1}{2}\left[\boldsymbol{y}_o - \mathcal{H}\left(\boldsymbol{x}\right)\right]^{\mathrm{T}}\mathbf{R}^{-1}\left[\boldsymbol{y}_o - \mathcal{H}\left(\boldsymbol{x}\right)\right],$$

(1)





in which, $\boldsymbol{x}_b$ is the background state, $\boldsymbol{y}_o$ the observation vector, $\mathcal{H}$ the nonlinear observation operator, and $\mathbf{B}$ and $\mathbf{R}$ are the
background and the observation error covariance matrix, respectively. $\mathbf{B}$ determines the error of the model at the observation
point, propagates the information from the observation in both horizontal and vertical directions and influences other variables.
To reduce the computational cost, the formulation used to find the solution from $J$ is incremental i.e., using $\delta x$ the difference
between the atmosphere state and the one from the background (Courtier et al., 1998).

AROME uses a 3DVar data assimilation scheme, with an hourly data assimilation cycle, to keep as close as possible to the
true state of the atmosphere (Brousseau et al., 2016; Gustafsson et al., 2018). The 3DVar analysis done at time $t_0$ considers all
observations between $t_0 - 30\,\mathrm{min}$ and $t_0 + 30\,\mathrm{min}$ to be valid at $t_0$. To avoid redundancy issues, only the observation closest
to $t_0$ is taken into account for stationary stations (PWS and SWS except ships) providing multiple observations during the
assimilation window. All available observations are submitted to a quality control procedure called screening, which removes
the observations deemed questionable. One of the screening steps compares the observation vector $\boldsymbol{y}_o$ to the background state
$\boldsymbol{x}_b$ using the observation operator $\mathcal{H}$, and keeps observations such as $\|\boldsymbol{y}_o - \mathcal{H}(\boldsymbol{x_b})\| < 5\sigma$ with $\sigma = \sqrt{\sigma_o^2 + \sigma_b^2}\sigma_{\mathrm{coef}}$, given $\sigma_o$
and $\sigma_b$, the standard deviation of observation and background errors, respectively, and $\sigma_{\mathrm{coef}} = 0.9^{-2}$ a supplementary factor
which artificially increases $\sigma$ allowing to keep more observations. Observations rejected by the screening are not used further.
For geopotential observations, $\sigma_o$ is set to $8g$, with $g = 9.806\,65\,\mathrm{m\,s^{-2}}$ the standard acceleration of gravity.

After this step, the observations kept are called active observations. This procedure matters, removing suspicious observa-
tions, which helps the convergence of the minimization algorithm and limits the spin-up period. However, some observations
might be rejected when they should not because of an inaccurate background. Additional quality control is tested in Sect. 3
to reject observations depending on the background departure of their time series rather than the background departure at one
specific time.

The 3DVar and 3DEnVar share the same formulation, with the only difference that $\mathbf{B}$ is static in the 3DVar and becomes
$\mathbf{B}_e$, an ensemble error covariance matrix, in the 3DEnVar (Michel and Brousseau, 2021). The matrix $\mathbf{B}_e$ is estimated for each
analysis from an Ensemble Data Assimilation (EDA) system composed of 50 members at $3.2\,\mathrm{km}$ horizontal grid spacing and
with 90 vertical levels (Brousseau et al., 2016). The advantage of the matrix $\mathbf{B}_e$ is to have a flow-dependent background error
statistics updated at each analysis, which allows the model to take better advantage of the observation. However, in order to
remove the sampling noise of the model error covariance estimate, a spatial localization is applied (Hamill et al., 2001), which
reduces the propagation of the innovation at a shorter spatial distance. The tuning of the horizontal localization has been found
to be optimal using a linear function from $25\,\mathrm{km}$ at the ground to $150\,\mathrm{km}$ at the highest model level (Michel and Brousseau,
2021). The vertical localization is set to about $0.15\,\mathrm{hPa}$ (Michel and Brousseau, 2021).

Observations of surface pressure from SWSs are assimilated in the form of a geopotential. Indeed, all assimilated observation
data are composed of an observed value and a pressure level. In the case of the geopotential, the observed value is set to $g \times z$,
with $z$ the altitude of the station. The model geopotential is retrieved using an observation operator described by Courtier et al.
(1998). The geopotential bias can be seen as a height bias by dividing the geopotential by the standard acceleration of gravity.





Surface pressure and geopotential variables are both used in this study. The surface pressure will be used when studying the observations, whereas the corresponding geopotential at a given altitude calculated by the observation operator of the DA scheme will be the variable used when studying the assimilation of the surface pressure observations.

### 170  2.3  Evaluation tools

When assimilating new observations, the associated $\mathbf{R}$ matrix has to be specified. The allocated part of the $\mathbf{R}$ matrix for PWSs will have the same characteristics as the one for the SWSs, i.e. it is assumed to be diagonal, and the diagonal values are the prescribed observation errors. This assumption requires verifying that observation errors are not correlated and, if they are, using techniques such as thinning to reduce this correlation.

The difference between an observation and its corresponding value in the model, called innovation, informs us of the concordance between both the observations and the model. Both the observation minus background (OmB) and the observation minus analysis (OmA) can be calculated using observation operators.

Desroziers et al. (2005) proposed a posterior diagnostic to estimate the observation error covariance matrix $\mathbf{R}$, which can be estimated by $E(\mathbf{OmA}[\mathbf{OmB}]^T)$. This method has some limitations. It assumes that the background and the observation error

covariance matrices are accurately specified and uncorrelated with a sufficiently large and representative sample of forecast error statistics (Pourret et al., 2022). However, the $\mathbf{B}$ matrix is not totally estimated or stored but specified through operators in the 3DVar scheme (Seity et al., 2011; Brousseau et al., 2014). Some limitations will be discussed further.

The estimation of the correlation between observations errors matters. Liu and Rabier (2003) showed that increasing the number of observations with uncorrelated errors generally increases the quality of the analysis and the forecast. However, if the

observations have correlated errors greater than a threshold (e.g., 0.2 as used by Liu and Rabier, 2003), it decreases the quality of the analysis and the forecast. To prevent observation errors from being spatially correlated, a thinning technique which removes observations at an optimal spatial resolution can be applied (Benáček et al., 2016). Thinning is done for observing systems providing high spatial resolution observations which have correlated observation errors, such as radar (Wattrelot et al., 2012), satellites (Dando et al., 2007), or aircraft (Pourret et al., 2021).

This method is not used for SWS, but given the high spatial density of the PWS (Fig. 1b), is tested further. To find this optimal spatial resolution, the Desroziers posterior diagnostic can be extended to a method estimating the spatial correlation of observations covariance error (Mile et al., 2019). First, for each distance bin, from 0 to 30 km at 1 km intervals, it finds equidistant pairs of stations. Then for each pair $(i, j)$ of stations in each bin, the covariate $\mathrm{OmA}_i.\mathrm{OmB}_j$ is calculated. Finally, the average of the covariates is normalized by the average of the first covariate, when the distance is zero, which corresponds

to the actual Desroziers diagnostic. This diagnostic gives an estimation of the spatial correlation errors, which can be used to estimate a minimum thinning length where the correlation errors are significantly low. It has the same limitation as the Desroziers diagnostic previously described.





**Table 1.** Overview of the experiments. 3DVar, 3DVarP, 3DEnVar and 3DEnVarP experiments are launched on 6 September 2021 00:00 UTC. The Monitor experiment is launched on 6 August 2021 00:00 UTC (one month earlier). The 3DEnVarP7UTC experiment is launched on 14 September 2021 07:00 UTC. BC means bias correction and QC means quality control.

| Experiments | Duration | Cycling | Data assimilation scheme | Use of PWS surface pressure observations |
|---|---|---|---|---|
| Monitor | 2 months | No | 3DVar | Raw observations monitored |
| 3DVar | | | 3DVar | No |
| 3DVarP | 1 month | Yes | | Assimilated after BC, QC, and thinning |
| 3DEnVar | | | 3DEnVar | No |
| 3DEnVarP | | | | Assimilated after BC, QC, and thinning |
| 3DEnVarP7UTC | 1 h | No | 3DEnVar | Assimilated after BC, QC, and thinning |

## 2.4 Design of the experiments

In order to evaluate the benefits of the new observations, the Observing System Experiments (OSE) framework is used to
compare a reference experiment with an experiment where new observations are assimilated (e.g., Andersson et al., 1991; Pourret et al., 2022). Making OSEs allows us to compare the background (i.e., 1 h forecast) or the analysis of both experiments to observations. In AROME-France the examination can be both at the surface with observations from stations and at different levels of the atmosphere with observations from radiosondes, aircraft, and satellites.

The Sect. 3 will use a monitoring experiment assimilating the raw pressure observations from the PWSs in a non-cycled
3DVar DA scheme. Then, the Sect. 4 will focus on long cycled experiments with the 3DVar and the 3DEnVar assimilation schemes. Finally, the Sect. 5 will use long, cycled experiments, and an extra experiment assimilating the new observations only at 07:00 UTC on 14 September 2021. All the experiments are described in Table 1. The experiments without assimilating PWS observations will also be called control experiments.

## 3 Pre-processing algorithm

All PWS observations are subject to a pre-processing algorithm before assimilation composed of a bias correction, quality control, and spatial and temporal thinning. The number of observations remaining after each step of the process is provided in Table 2. Starting from raw observations, they are interpolated at round hours, which concomitantly removes observations that are too sparse. Stations with identical coordinates are assumed to have not been properly set up by the owner, leading to automatic placement based on their wireless IP address (Meier et al., 2017). These stations are further removed.

### 3.1 Bias correction and quality control

Figure 2a shows one-month time series of surface pressure observations from 30 PWSs close to the city of Toulouse (i.e., less than 0.05 ° from the city centre), located in the south of France. Most of the time series exhibit concomitantly similar pressure




**Table 2.** Average number of PWS surface pressure observations following the pre-processing process between 6 September 2021 00:00 UTC and 5 October 2021 23:00 UTC (717 time steps). Each step of the process is described in the text. Std refers to standard deviation of the OmBs last rolling month (see text).

|  | Raw | Preparation | | BC | QC | | |
|---|---|---|---|---|---|---|---|
|  |  | interpolation | Same lat-lon | >1/2 of the period | Station has moved | >1/4 during the last 48 h | std > 60 $m^2$ $s^{-2}$ |
| PWS remaining | 55187 | 52334 | 50757 | 48275 | 47748 | 47564 | 46800 |
| % of total PWS | 100 % | 94.8 % | 92.0 % | 87.4 % | 87.0 % | 86.2 % | 84.8 % |

variations, and large differences in the mean value. However, the area being relatively flat (i.e., less than 150 m from higher to lower point in this area), it is not realistic to have a systematic difference up to 25 hPa (approx. 250 m), i.e., a first glance

of the different systematic errors at each station location. Then, it has to be confirmed with long-term statistics, which can be done with the monitoring experiment. In this experiment, which assimilated the raw pressure observations from the PWSs, the observations are compared to the model background (i.e. the one-hour forecast from the operational AROME model), giving the OmB for the whole set of observations. The 30 PWS time series have systematic biases from $-500 \, m^2 \, s^{-2}$ to $1200 \, m^2 \, s^{-2}$ in terms of geopotential (Figure 2b). More globally, the Figure 3 shows the whole set of geopotential OmB during the pre-

processing. The raw data exhibit the biases from $\pm 1000 \, m^2 \, s^{-2}$ (looking at the whiskers, representing the percentile range of the data, i.e. from 5th to 95th percentiles). Therefore, a systematic bias correction is crucial.

To implement a geopotential bias correction for PWS data based on AROME, the previously described monitoring experiment with a non-cycled 3DVar DA scheme has been used. PWS individual biases are then obtained by averaging the geopotential OmB for each station over a rolling time window of 1 month to prevent the correction from depending on the me-

teorological situation and for robust statistics. This bias is then subtracted from the actual observation which, at the end, only changes the altitude of the station in the assimilation scheme. PWSs with missing observations for more than half the rolling time window are eliminated to ensure the robustness of the bias. This bias correction reduces the bias range to $\pm 15 \, m^2 \, s^{-2}$ and removes 4.6 % more stations while keeping numerous ones with heterogeneous quality (Fig. 3b). Contrary to SWSs, some PWSs can suddenly move, and their spatial coordinates or altitude metadata change. Some other stations also show sudden

pressure change without an abnormal change in their metadata (Fig. 2a). That is why a supplementary quality control has been tested. First, PWSs that have moved from more than 100 m on the horizontal and 1 m on the vertical are discarded. That represents 0.4 % more stations removed. Also, an interruption in the PWS observation time series is suspected to be an early sign of erroneous observations: PWS with missing observations for more than a quarter of the last 48 h are eliminated. Finally, PWSs whose geopotential observations deviate too much from AROME background, i.e. with a standard deviation of their last

month's geopotential OmB above, $60 \, m^2 \, s^{-2}$ are removed. This empirical threshold removes the tail of the OmB distribution (i.e., 1.4 % of the observations) corresponding to unrealistic observations, while keeping physical patterns. In the end, 84.8 % of the stations are kept, and the range of the geopotential OmB is under the SWS ones (Figs. 3c, d).

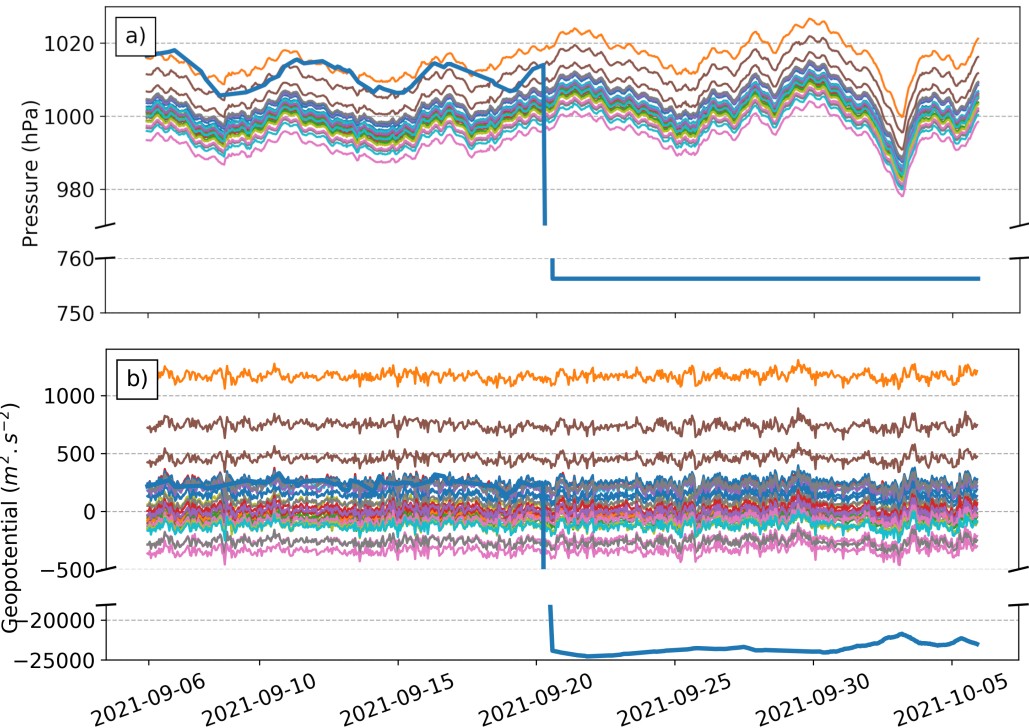

**Figure 2.** Time series of (a) surface pressure observations and (b) corresponding geopotential OmBs from 30 PWSs around Toulouse and one anomalous PWS during the study period from 6 September to 5 October 2021. The background is from the Monitor experiment.

## 3.2 Thinning

After the bias correction and the quality control, the number of PWS observations within a 1.3 km horizontal grid cell of
AROME ranges from 0 to 69, 69 being reached in the centre of Paris. In order to keep a minimal weight to the model's background and other observations at such grid points where several PWS geopotential observations are available, it is chosen to thin PWS observations at 1.3 km, the model's horizontal grid spacing, for the whole studied period, independently of the assimilation scheme. Among PWS observations available in a grid point, 1 is randomly drawn. Such thinning, generally never applied for independent surface observations, is generally applied in DA to minimize the risk of having correlated observation
errors. To verify that PWS geopotential observation errors after thinning are not correlated, a spatial Desroziers diagnostic (Sect. 2.3) is computed for two analyses. The analysis have been chosen at midnight and at mid-afternoon.

Figure 4 presents this diagnostic for pre-processed pressure observations from PWSs. From the first bin (1 km) there is limited spatial correlation of observation error. This diagnostic shows small variability between the two assimilation schemes and the hour of study. It confirms that, if a thinning radius is chosen, there is no need to increase the thinning radius above
1.3 km to mitigate the issue of correlated errors. The effect of assimilation without thinning or shorter thinning lengths is not


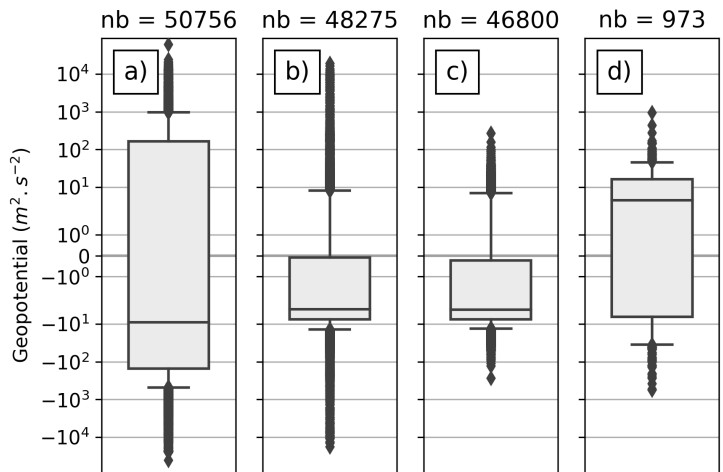

**Figure 3.** Box plots of geopotential OmB during the one-month study period. Geopotential observations originate from (a) PWS raw data, (b) PWS bias-corrected data, (c) PWS assimilated (bias-corrected and quality-controlled), and (d) SWS assimilated. The background fields are coming from the non-cycled monitoring experiment. Box-plot titles indicate the hourly average number of stations. The bar across the box represents the median, and its borders represent the 25th and 75th percentiles. The box-plot whiskers indicate the 5th and 95th percentiles of the data. Outliers are plotted as separate dots.

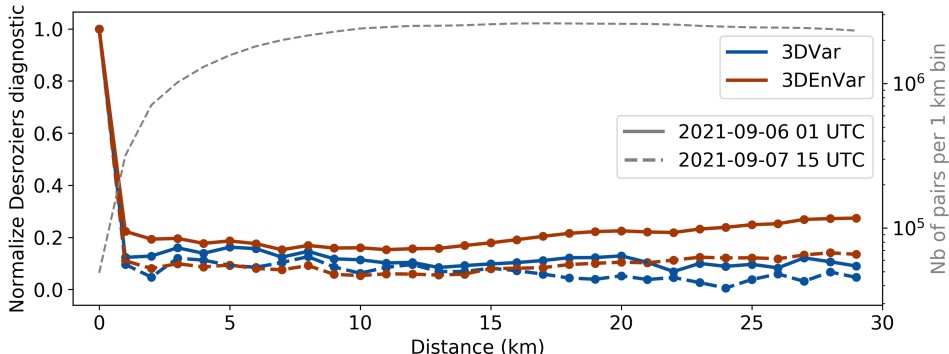

**Figure 4.** Spatial Desroziers diagnostic of PWS surface pressure observations. The colours represent the assimilation scheme, with the 3DVar, in blue, and 3DEnVar, in brown. The diagnostic is done at (plain lines) 01:00 UTC on 5 September 2021 and (dashed lines) at 15:00 UTC on 6 September 2021. In grey, the number of pairs of equidistant stations (in 1 km bins).

tested. After thinning, there is an average of 27 889 PWS surface pressure observations per hour assimilated, which represents 51 % of the initial amount of PWS observations.





**Table 3.** Mean and standard deviation of geopotential OmA and OmB ($m^2\,s^{-2}$) and Desroziers diagnostic ($\widetilde{\sigma_O}$ in $m^2\,s^{-2}$) for SWSs located in France and PWSs over the one-month study period, which represent respectively 212 and, 27889 observations per hour in average over the 717 hours of the experiments.

| | Monitor | | 3DVar | | 3DVarP | | | | 3DEnVar | | 3DEnVarP | | | |
|---|---|---|---|---|---|---|---|---|---|---|---|---|---|---|
| | PWS | | SWS | | SWS | | PWS | | SWS | | SWS | | PWS | |
| | mean | std | mean | std | mean | std | mean | std | mean | std | mean | std | mean | std |
| OmA | n.a. | n.a. | 4.7 | 26.5 | 11.9 | 25.4 | −0.1 | 14.6 | 12.8 | 38.0 | 12.2 | 24.8 | 0.2 | 15.9 |
| OmB | -2.6 | 36.3 | 11.4 | 43.3 | 16.7 | 41.7 | 3.9 | 48.9 | 15.6 | 46.4 | 15.3 | 41.3 | 2.6 | 49.2 |
| $\widetilde{\sigma_O}$ | n.a. | | 31.2 | | 27.7 | | 15.7 | | 43.6 | | 29.3 | | 19.7 | |

## 4   Results of 1 month assimilation experiments

When statistical values (STAT) such as standard deviation or root-mean-square (RMS) of time series are compared between experiments, their relative evolution in an experiment (XP) with respect to another (CTRL) is given by:

$$\Delta\text{STAT} = \frac{\text{STAT}_{\text{XP}} - \text{STAT}_{\text{CTRL}}}{\text{STAT}_{\text{CTRL}}} \tag{2}$$

Special attention will be paid when the CTRL statistical value is close or equal to 0, in which case the relative evolution is not defined. The significance of the values is determined using a bootstrap method, where the RMS is computed from the time series sampled with replacement 1000 times. The 1000 RMS values form a distribution, from which 95 % confidence intervals are estimated using the "percentile" method (Virtanen et al., 2020).

### 4.1   Impact on geopotential analyses and 1 h forecasts

Statistics of geopotential OmA and OmB for SWSs and PWSs gathered in Table 3 reveal the ability of the assimilation schemes to take advantage of the PWS geopotential observations.

In the 3DVar experiment, the OmA geopotential mean for SWSs is equal to $4.7\,m^2\,s^{-2}$, which shows a little bias between AROME analyses and the SWS geopotential observations over France. Over the whole domain, it decreases to $2.0\,m^2\,s^{-2}$ (not shown). Whereas, in the 3DEnVar experiment, the OmA geopotential mean for SWS is higher and equal to $12.8\,m^2\,s^{-2}$. The geopotential standard deviation OmA increases by 43 % from 26.5 to $38.0\,m^2\,s^{-2}$ between 3DVar and 3DEnVar. This can be due to the finer localization given to geopotential SWS observations in 3DEnVar than in 3DVar. In 3DVarP and the 3DEnVarP, experiments in which PWS geopotential observations are assimilated, the OmA geopotential means for PWSs are closer to zero (respectively $−0.1\,m^2\,s^{-2}$ and $0.2\,m^2\,s^{-2}$). This result was expected given the prior PWS bias correction using AROME's OmB statistics over a rolling month.

Between 3DVar and 3DVarP, the OmA geopotential mean for SWSs increases from 4.7 to $11.9\,m^2\,s^{-2}$, while the geopotential standard deviation OmA for SWSs decreases by 4 %. Similar results are found when observations are compared to the back-





ground: the OmB geopotential mean for SWSs increases, while the geopotential standard deviation OmB for SWSs decreases
by 4 %. Time series of geopotential mean and standard deviation OmB shows that these results are the same every day during
the 1-month period studied (Fig. 5a), i.e. the OmA geopotential mean (resp. standard deviation) for SWSs is systematically
increased (resp. decreased).

Between 3DEnVar and 3DEnVarP, the OmA geopotential mean for SWSs decreases from 12.8 to 12.2 $\mathrm{m^2\,s^{-2}}$, while the
geopotential standard deviation OmA for SWSs decreases by 53 %. Similar trends are found when observations are compared
to the background: the OmB geopotential mean for SWSs decreases, while the geopotential standard deviation OmB for
SWSs decreases by 12 %. Time series of geopotential standard deviation OmB shows that these trends are the same every
day during the 1-month period studied (Fig. 5b); which is not systematic for geopotential mean OmB. Times series also show
that improvements in OmB statistics (mean and standard deviation) of 3DEnVarP compared to 3DEnVar are larger on days
when 3DEnVar OmB statistics are furthest from zero, which corresponds to days of perturbed weather, such as on 8 and 14
September and 5 October 2021.

In the framework of these experiments, considering SWS observations as the anchoring reference, the assimilation of PWS
observations in a 3DVar scheme has a mixed impact: it increases OmA and OmB geopotential mean but decreases OmA
and OmB geopotential standard deviation during the one-month study period. The large increase in OmA geopotential mean
when PWS observations are assimilated could be explained by the PWS bias correction method, which uses OmB and not
OmA statistics for it. On the contrary, the assimilation of PWS observations in a 3DEnVar scheme systematically improves all
statistics of both analyses and 1 h forecasts.

The Desroziers diagnostic ($\widetilde{\sigma_O}$), described in Sect. 2.3, calculated from OmA and OmB statistics, encompass these results.
For all experiments, values of this diagnostic estimating the observation error range from 15.7 to 43.6 $\mathrm{m^2\,s^{-2}}$ (Table 3), which is
only between 20 and 56 % of the prescribed value in AROME (78.4532 $\mathrm{m^2\,s^{-2}}$). For SWSs, the Desroziers diagnostic increases
by 40 % from 31.2 to 43.6 $\mathrm{m^2\,s^{-2}}$ between 3DVar and 3DEnVar experiments, which shows a high sensitivity to the choice of
the assimilation scheme. Also, the Desroziers diagnostic varies for SWSs when PWSs are assimilated: it decreases by 12 %
between 3DVar and 3DVarP and by 48 % between 3DEnVar and 3DEnVarP. Finally, the PWS bias correction explains why
the Desroziers diagnostic is lower for PWSs than for SWSs in 3DVarP and 3DEnVarP experiments, which could be wrongly
interpreted as PWS geopotential observations are of better quality than SWS geopotential observations (see Appendix A).
These findings show some limits of this diagnostic, which should be used with caution, especially when its assumptions such
as that the average OmB and OmA are close to zero are not fulfilled.

### 4.2    Impact on 1 h forecasts of surface pressure, temperature, relative humidity and wind

To show how the background (1 h forecast) deviates from SWS observations, the RMS OmB of a given variable observed near
the ground by SWSs is calculated. This score is presented in Table 4.

The comparison of 3DEnVar with respect to 3DVar shows an 8.0 % increase in RMS OmB of surface pressure for SWSs
over the AROME domain and 9.3 % over France. It indicates that the 3DEnVar background deviates more than the 3DVar
background from SWS observations of surface pressure. For 2 m temperature and 10 m zonal and meridian wind, increases in

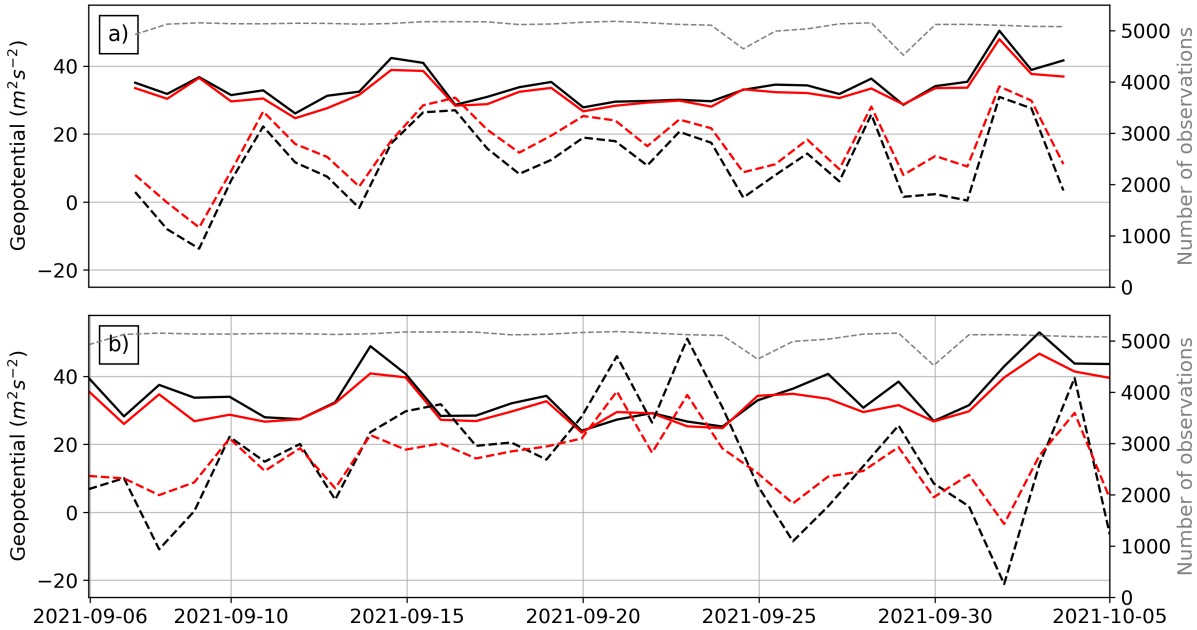

**Figure 5.** Daily average geopotential (dashed line) mean and (plain line) standard deviation OmB for (a, black) 3DVar, (a, red) 3DVarP, (b, black) 3DEnVar, and (b, red) 3DEnVarP experiments over the one-month study period. Only observations available for all experiments are taken. The grey dashed line is the daily number of observations.

**Table 4.** ΔRMS OmB of surface pressure, 2 m temperature, 2 m relative humidity and 10 m zonal and meridian wind for SWSs over the one-month study period. Negative values of an experiment (XP) w.r.t. another (CTRL) indicate that 1 h forecasts of XP are closer than CTRL to SWS observations. Significant values are in bold (see text for details).

| | Surface pressure | | Temperature | | Relative humidity | | Zonal wind (U) | | Meridional wind (V) | |
|---|---|---|---|---|---|---|---|---|---|---|
| | AROME domain | France | AROME domain | France | AROME domain | France | AROME domain | France | AROME domain | France |
| 3DEnVar w.r.t. 3DVar | **8.0 %** | **9.3 %** | **2.2 %** | **1.2 %** | **3.6 %** | **−1.1 %** | **2.0 %** | **1.1 %** | **2.0 %** | **1.3 %** |
| 3DVarP w.r.t. 3DVar | 0.8 % | 0.7 % | 0.0 % | −0.8 % | 0.0 % | 0.0 % | −1.4 % | **−1.6 %** | −1.4 % | **−1.5 %** |
| 3DEnVarP w.r.t. 3DEnVar | **−3.0 %** | **−10.3 %** | **0.6 %** | **0.8 %** | **0.8 %** | **1.3 %** | 0.2 % | 0.0 % | -0.2 % | -0.2 % |

RMS OmB range from 1.1 % to 2.0 %. Only the RMS OmB of 2 m relative humidity decreases by 1.1 % for SWSs over France but still increases by 3.6 % over the AROME domain.

The comparison of 3DVarP with respect to 3DVar shows slightly positive ΔRMS OmB of surface pressure for SWSs over France (0.7 %) and the AROME domain (0.8 %), showing a slight degradation, but not statistically significant. Despite this, a significant decrease of −1.6 % and −1.5 % of RMS OmB of the zonal and meridional wind for SWSs located in France are found.



The comparison of 3DEnVarP with respect to 3DEnVar shows significantly negative ΔRMS OmB of surface pressure for
SWSs, reaching −10.3 % over France and −3.0 % over the AROME domain. Small but significant degradation from 0.8 % to
1.3 % is found on 2 m temperature and 2 m relative humidity.

These experiments show that PWS assimilation improves significantly the quality of the surface pressure background when a
3DEnVar scheme is used, and partly compensates for the significant degradation associated with the transition from a 3DEnVar
system to the 3DVar. Even if they are statistically significant, only a little improvement of the background is found in the
AROME 3DVar scheme in terms of wind when PWS geopotential observations are assimilated, with a neutral impact on
surface pressure.

## 4.3  Impact on the forecasts up to 30 h range

For 3DVar, 3DVarP, 3DEnVar, and 3DEnVarP experiments, forecasts starting from the analyses at 00:00, 06:00, 12:00 and
18:00 UTC were run each day during the one-month study period. Statistics of observation minus forecast (OmF) and their
evolution between experiments (as indicated in Eq. 2) are computed. By definition, OmF at 0 h time step is equal to OmA but
OmF at 1 h time step is not equal to OmB because of the use of the incremental analysis update technique in AROME-France
forecasts (Brousseau et al., 2016).

At the analysis (0 h), ΔRMS OmF of MSLP for 3DVarP w.r.t. 3DVar is negative and equal to −7.5 %, then rapidly increases
during the first hours of the forecast and is positive after only 3 h of forecast (Fig. 6a). For 3DEnVarP w.r.t. 3DEnVar, ΔRMS
OmF of MSLP is negative and equal to −24 % at the analysis, also rapidly increases during the first hours of the forecast, but
remains negative before 12 h of forecast. This shows that the benefit of the assimilation of PWS observations quickly disappears
in 3DVarP, while it remains in the 3DEnVarP experiment. Small but significant degradation is noticeable between 15 to 21 h of
forecast for 3DVarP w.r.t. 3DVar, which is less pronounced and not significant for 3DEnVarP w.r.t. 3DEnVar. The influence of
the assimilation of PWS observations on other variables near the ground is less pronounced, and no significant results are seen
(not shown).

Impact at different levels of the atmosphere can be assessed with the help of radiosoundings (Fig. 6b). Because PWS ob-
servations are only assimilated over France or near its borders, only 5 French radiosoundings launched twice a day are taken
into consideration. Over the one-month study period, it represents between 220 (at 1000 hPa) and 272 observations. Negative
ΔRMS OmA of geopotential near the ground and for the whole troposphere is seen, reaching at 1000 hPa −11 % for 3DVarP
w.r.t. 3DVar and −24 % for 3DEnVarP w.r.t 3DEnVar. This shows a substantial improvement of geopotential in the analysis
throughout the troposphere when PWS observations are assimilated. Regarding other variables, improvement is seen on the
wind heterogeneously for the whole troposphere for 3DVarP w.r.t. 3DVar experiment (not shown), and no significant change
could be seen for other radiosounding parameters.

AROME-France being the finer-scale operational model available over France, one of its added values lies in the forecasting
of local precipitation events. One common verification score called AROME index (also called IP16) described in Amodei et al.
(2015) synthesises regional Brier skill scores (BSS_NO) of wind gusts and 6 h-accumulated rainfall up to 24 h of forecast. It
is computed for the different experiments and is shown in Figure 7. IP16 increases from 0.797 to 0.827 between 3DVar and



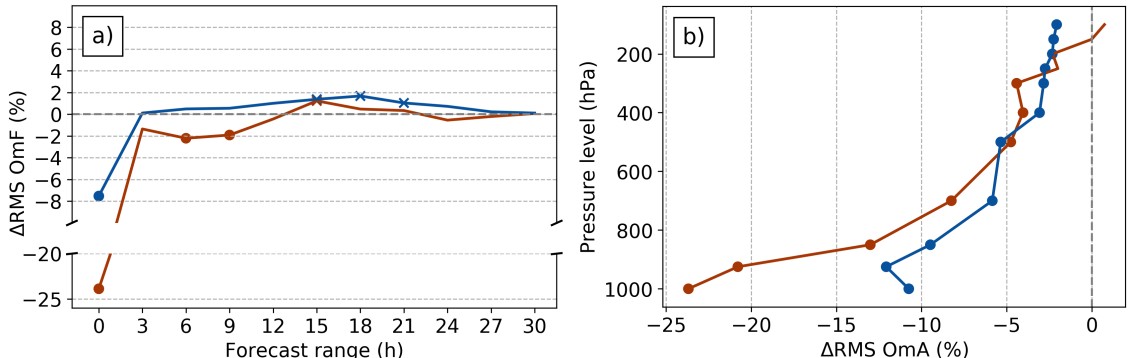

**Figure 6.** (a) ΔRMS OmF of MSLP for SWSs over France between 0 and 30 h forecast range and (b) ΔRMS OmA of geopotential for radiosoundings launched over France for (blue) 3DVarP w.r.t. 3DVar and (brown) 3DEnVarP w.r.t. 3DEnVar. Points (respectively crosses) indicate improvement (resp. degradation) at 90 % statistical significance level.

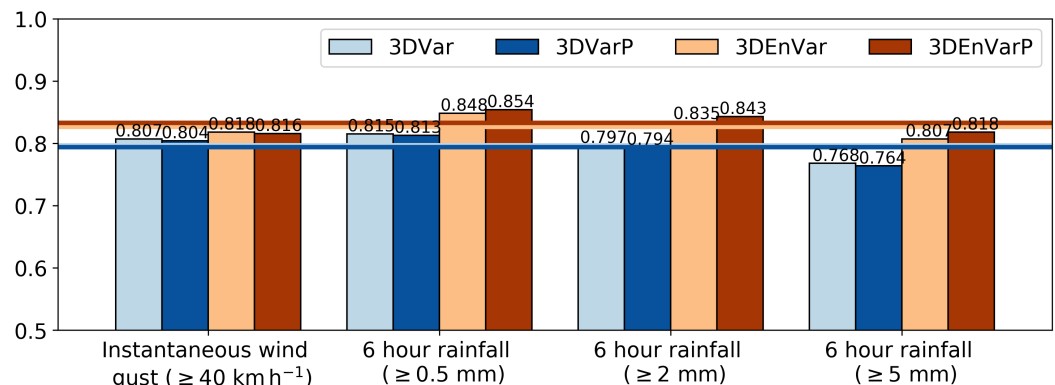

**Figure 7.** IP16 score of forecasts launched at 00:00 UTC during the one-month study period except the first day (7 September to 5 October 2021) averaged over 4 forecast times (6 h, 12 h, 18 h, 24 h) given by horizontal bars for 3DVar (light blue), 3DVarP (navy blue), 3DEnVar (beige), 3DEnVarP (brown) experiments. All columns correspond to regional Brier skill scores (with a spatial window of 50 km) contributing to IP16: instantaneous wind gusts above 40 km h$^{-1}$, 6 h-accumulated rainfall above or equal to 0.5 mm, 2 mm, and 5 mm. Higher is better.

3DEnVar, which is a 3.8 % improvement associated with the change of the DA scheme. The assimilation of PWS geopotential observations decreases IP16 by −0.3 % in 3DVarP with respect to 3DVar, and increases by 0.7 % in 3DEnVarP with respect to 3DEnVar.

When looking in detail at BSS_NO composing the IP16, the largest improvement of BSS in 3DEnVarP w.r.t. 3DEnVar is found for the highest threshold of accumulated rainfall: BSS_NO increases by 0.7 % for the 0.5 mm threshold against 1.3 % for the 5 mm threshold. Improvements in BSS_NO are found particularly between 6 to 18 h of the forecast, and vanish after 24 h (not shown). Results concerning the 14 September event are developed in the next section.




## 5  Results on the 14 September 2021 heavy precipitation event

### 5.1  Presentation of the case

The Mediterranean region is regularly affected by heavy precipitation events (HPEs, Doswell et al., 1998; Nuissier et al., 2020; Caumont et al., 2021). On 14 September 2021, a mesoscale convective system (MCS) remained quasi-stationary for 8 h, dumping up to 260 mm of rain in 3 h between 07:15 and 10:15 UTC in the city of Saint-Dionisy, in the Gard department (south of France). The particular organization of mesoscale vertical circulations, clouds, hydrometeors, and precipitation inside MCSs can lead to the formation of surface pressure structures which can be observed from the ground.

When PWS surface pressure observations — quality-controlled by MC20 algorithm — and SWS surface pressure observations are combined, surface pressure structures appear near and below the MCS (Fig. 8). At 07:00 UTC a low-pressure anomaly in front of the precipitation system is observed, which could be associated with updrafts, increased relative humidity, and increased temperature; and at the rear high-pressure anomaly, potentially due to downdrafts. These surface pressure anomalies could not be seen with SWS observations only: therefore, this section examines the impact of assimilating PWS observations of pressure in the area where these anomalies are observed.

### 5.2  Impact of PWS observations assimilation on surface pressure increments

Figure 9 shows increments of surface pressure for experiments assimilating PWS observations of geopotential (3DVarP, 3DEnVarP) or not (3DVar, 3DEnVar). Because the four experiments are cycled and starting from much earlier, they have a different background at 07:00 UTC. As the surface pressure increments are not to be compared two by two, only the ability of the different experiment configuration to take into account the observations is emphasised. Other observations are assimilated in this area at 07:00 UTC, coming from SWSs, from radars and from an aircraft flying towards Marseille-Provence airport (not shown).

Increments in experiments using a 3DVar DA scheme (Fig. 9a, b) are smoother than increments in experiments using a 3DEnVar DA scheme (Fig. 9c, d). These differences are due to the differences in background error covariance matrices between both schemes. In the 3DVar experiment (Fig. 9a), all pressure observations from SWSs are consistent with the increment signs, which indicate that with their large analysis increment, they are more predominant than other assimilated observations. This is not the case for the 3DEnVar experiment (Fig. 9c), where anomalies appear close to the centre of the MCS. They are not consistent with the 2 closest SWS surface pressure observations. To know from which observations these anomalies were coming, a supplementary experiment has been done rejecting the radar which is the other type of observation assimilated close to the anomalies; which effectively removes the anomalies (not shown). This confirms that surface pressure observations have a lower impact than other assimilated observations (here the radars) in this specific case with the 3DEnVar scheme.

When assimilating PWS observations, the two assimilation schemes react differently. For the 3DVarP (Fig. 9b), all the positive innovations from PWSs add up to form a large positive anomaly. A small group of observations, close to the MCS, are not consistent with this anomaly, which shows the difficulties that the 3DVar DA scheme has to take advantage of the local scale information. However, the 3DEnVarP experiment (Fig. 9d) effectively achieves to push the model according to the


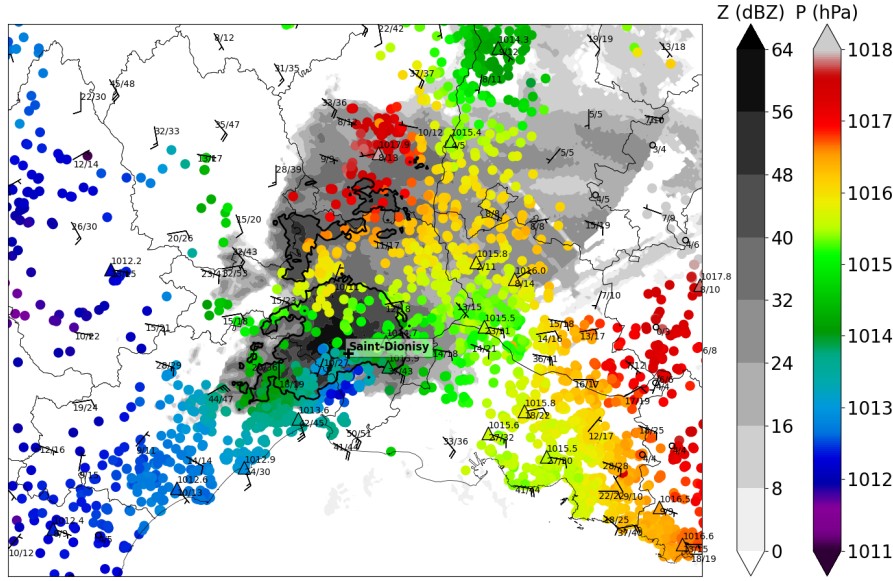

**Figure 8.** MSLP observations of SWS and PWS networks at 07:00 UTC on 14 September. SWSs are indicated by coloured triangles with black contours, and PWSs by coloured circles. The instantaneous (500 or 250 ms) wind gust during the last minute is shown with barbs (in kt). Two values separated by a slash indicate respectively from left to right the instantaneous wind gust during the last minute and during the last 10 min in $\mathrm{km\,h^{-1}}$. Base reflectivity (Z) in grey colours indicates thunderstorm activity and location. Reflectivities over 40 dBZ are illustrated with bold black contours.

innovations brought by PWS observations. The number of PWS observations assimilated compensate for the small localisation given to the surface observations in the 3DEnVar DA scheme. Only a few stations remain at the boundary between positive and negative anomalies, which do not correspond to the increment. As 3DEnVarP is the only experiment able to reproduce the local anomaly of surface pressure observed by PWSs in the 07:00 UTC analysis, the following section will compare experiments with a 3DEnVar DA scheme.

### 5.3 Impact on forecasts from a single hour PWS assimilation

Once known the local increments of pressure added by the PWSs with the 3DEnVar scheme, the influence on other variables is explored. In that way, an experiment using the 3DEnVar DA scheme and assimilating PWS observations only at 07:00 UTC was realised (Fig. 10). It is called hereafter 3DEnVarP7UTC. In the AROME DA system, the surface pressure, temperature, specific humidity, and the two components of the wind are analysed every hour, while the other prognostic model fields, such as vertical divergence, are only updated during the forecast steps of the assimilation cycles. In consequence, with the help of the 3DEnVar background covariance error matrix, the PWS geopotential observations induce surface pressure, temperature, and relative humidity changes close to the surface. Indeed, above the positive surface pressure increments, between 3 and 4 °E,

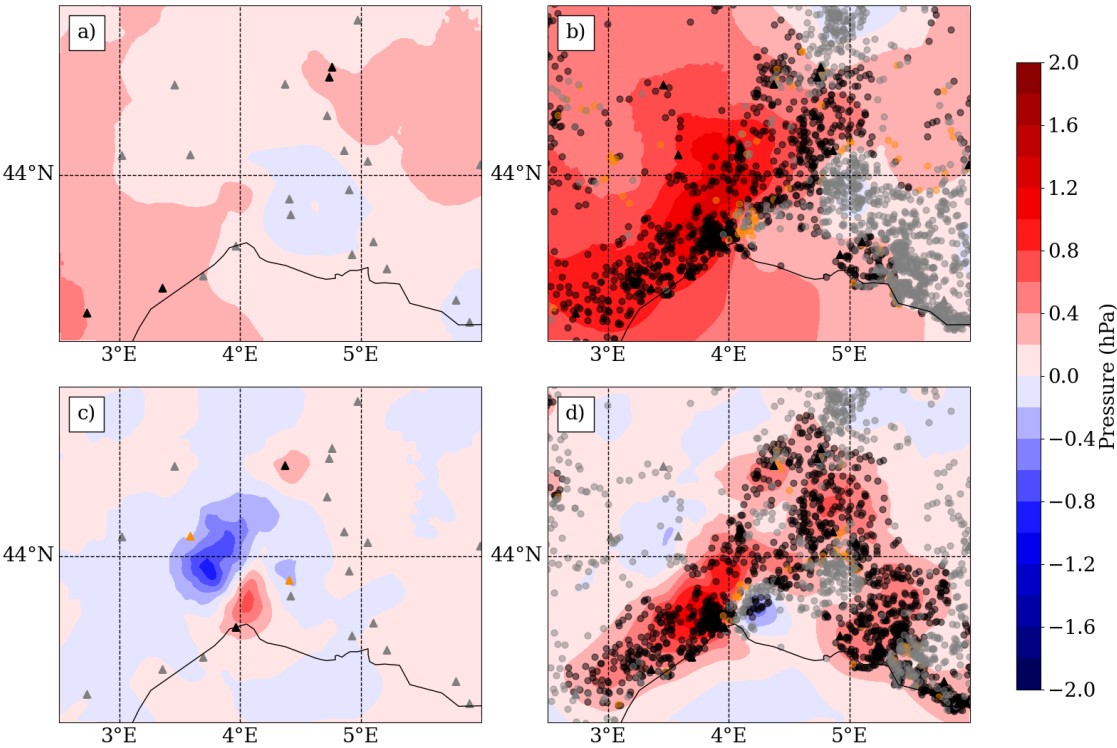

**Figure 9.** Surface pressure increments (analysis minus background) in (a) 3DVar, (b) 3DVarP, (c) 3DEnVar, and (d) 3DEnVarP experiments on 14 September 2021 at 07:00 UTC. Locations of the stations whose observations of surface pressure are assimilated are indicated by triangles for SWSs and circles for PWSs. Colours of triangles and circles indicate for each surface pressure observation the change in analysis departure (OmA) with respect to background departure (OmB): reduced in black such as $|\mathrm{OmA}| - |\mathrm{OmB}| < -0.2\,\mathrm{hPa}$, increased in orange such as $|\mathrm{OmA}| - |\mathrm{OmB}| > 0.2\,\mathrm{hPa}$ or neutral in grey such as $||\mathrm{OmA}| - |\mathrm{OmB}|| < 0.2\,\mathrm{hPa}$.

a negative temperature difference implying a denser air mass appears (Fig. 10b, c). Vertical velocity, being a diagnostic field, is not updated at the analysis and only small anomalies appear due to changes in surface pressure and thus pressure of all grid points on the vertical (Fig. 10d). Regarding the evolution of these anomalies, after a few time steps of forecast, the AROME model balances the information provided by the surface pressure increments into other prognostic fields, and uses them to

create new meteorological structures. Spatial size of differences in surface pressure in Fig. 10e is lower compared to the size of the increments visible at the analysis. Near 4 °E, after 50 min of forecast, a dipole of positive and negative vertical velocity differences appears (Fig. 10h). At that time, multiple fine-scale differences, up to 1 °C in temperature and up to 20 % in relative humidity shows that the location of convective cells inside the MCS is modified. This particular event shows that changes in surface pressure have an effect on the other variables in the model which can rapidly extend all the way to the top of the

troposphere. As such storms occur rapidly and over a small geographical area compared with the area of the model domain,

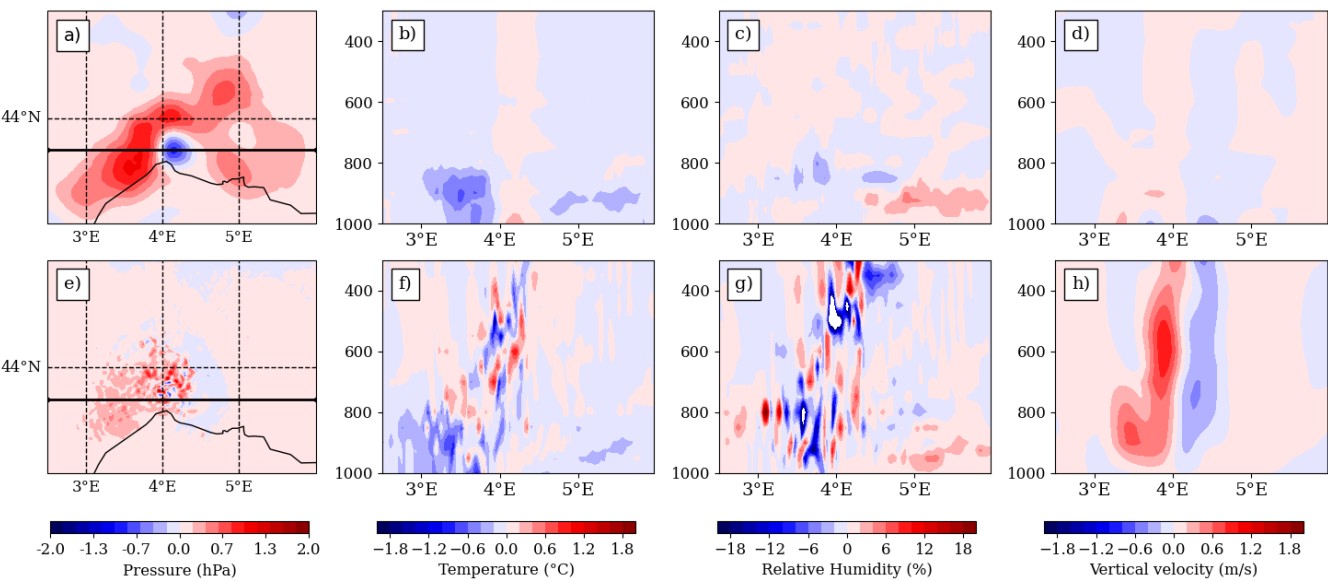

**Figure 10.** Differences between 3DEnVarP7UTC and 3DEnVar experiments of (a, e) surface pressure, (b, f) temperature, (c, g) relative humidity and (d, h) vertical velocity (a–d) at the analysis of 07:00 UTC and (e–h) at 50 min forecast time. (b–d, f–h) Differences along the vertical cross-section indicated by the thick black line in (a) in pressure coordinates (hPa).

the modifications of temperature and relative humidity seen in this figure are barely visible on assimilation experiment scores over a time period of 1 month shown in Sect. 4.

## 5.4 Impact on forecasted rainfall

The MCS remained quasi stationary in the southern part of the Gard department which caused heavy rainfall during the event.
The rainfall hourly time series analysed by ANTILOPE, the Météo-France operational precipitation analysis combining radar and rain gauge observations at 1 km grid spacing (Laurantin, 2008), shows the chronology of the event (Fig. 11d). Rainfall started over the area at 00:00 UTC, maximum hourly rainfall continuously increased until 09:00 UTC, and decreased after 09:00 UTC. Between 08:00 and 09:00 UTC, ANTILOPE analysis (Fig. 11c) shows two convective cells, the southern being the most intense with an hourly maximum rainfall accumulation of 144 mm. The corresponding 9 h forecasts starting from the
3DEnVar and 3DEnVarP analyses are shown in Fig. 11a and Fig. 11b. The 3DEnVar experiment forecasts rainfall over the entire Gard department and even north of it, whereas the 3DEnVarP experiment is closer to ANTILOPE for both the spatial extension and the location of the rainfall maximum in the south of the Gard department. Looking at the maximum rainfall accumulation, the 3DEnVarP experiment with 96 mm is closer to the observation (144 mm) than the 3DEnVar experiment, with 65 mm. It can be assumed that the vertical velocity anomalies previously observed could generate more intense precipitation rates. Fractions
Skill Scores (FSS, Mittermaier, 2018) show no advantage of 3DEnVarP w.r.t. 3DEnVar for small thresholds of precipitation (e.g., 1 mm). However, over 5 mm thresholds, the FSS is significantly better for the 3DEnVarP experiment (not shown). This

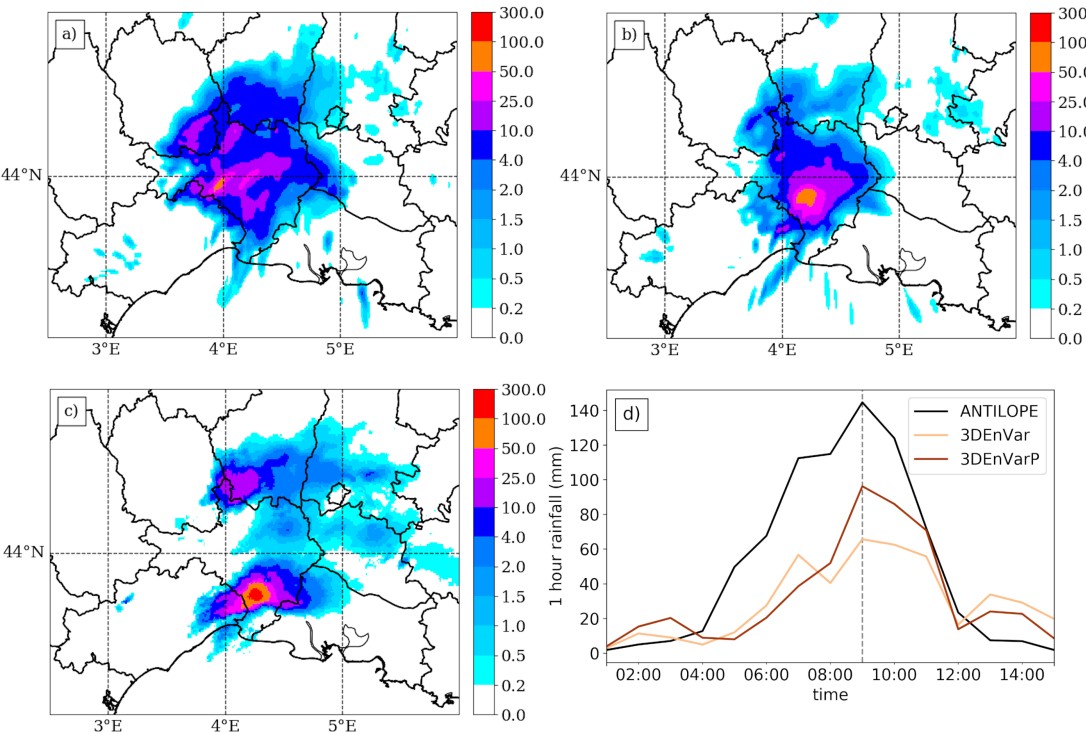

**Figure 11.** Rainfall accumulation (mm) on 14 September 2021 between 08:00 and 09:00 UTC of forecasts starting from the (a) 3DEnVar and (b) 3DEnVarP analyses at 08:00 UTC on 14 September 2021 and (c) analysed by ANTILOPE. (d) Time series of hourly maximum rainfall over the area of forecasts starting from the (light brown) 3DEnVar and (brown) 3DEnVarP analyses and (black) analysed by ANTILOPE.

means a better location and intensity of the hourly maximum rainfall accumulation of precipitation at this time. In addition, the time series of the hourly rainfall accumulation of the 3DEnVarP experiment is closer to ANTILOPE than the 3DEnVar experiment.

Finally, what forecasters are interested in is the anticipation of the event. Then, the added value of the 12 h forecast launched from the 3DEnVarP analysis at 00:00 UTC on 14 September 2021 with respect to the 12 h forecast launched from the 3DEn-Var analysis at the same time is explored. The maximum rainfall accumulation observed during this period is 320 mm. The maximum rainfall accumulation of the 12 h forecast launched from the 3DEnVarP analysis is 221 mm, which is 22 % higher than the 3DEnVar one (180 mm). Looking at the FSS, small but significant degradation is seen at the 2 mm threshold, whereas

significant improvement can be seen at higher thresholds, especially at 50 and 100 mm (Fig. 12). In this case, the FSS of the 12 h forecast at 100 mm threshold is significantly improved at a window size of 1° in latitude and longitude. This could be explained by more realistic analyses a few hours before the forecast launch, thanks to the assimilation of PWS surface pressure observations.





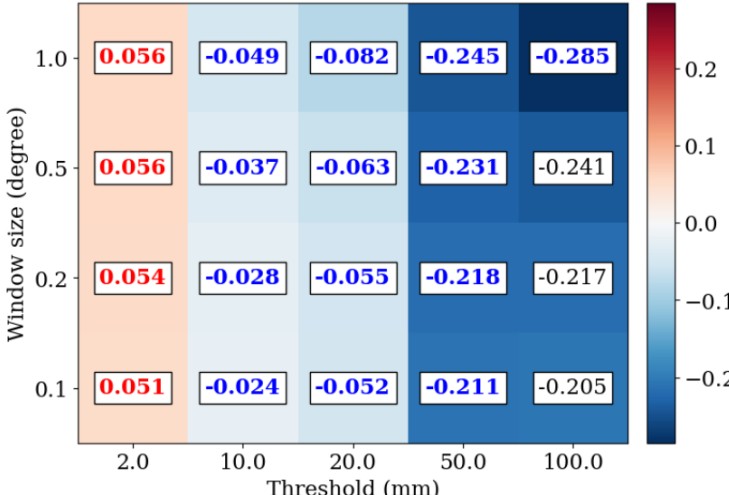

**Figure 12.** Difference in FSS of 12 h precipitation forecasts starting from the 3DEnVar and 3DEnVarP analyses at 00:00 UTC on 14 September 2021. Difference is computed as FSS(3DEnVarP)−FSS(3DEnVar). FSS is computed for 2, 10, 20, 50, and 100 mm thresholds and spatial neighbourhoods of 0.1, 0.2, 0.5, and 1° in latitude and longitude. Significant positive (resp. negative) values are in red (resp. blue).

## 6   Discussion and conclusion

This study has examined the impact of the assimilation of spatially dense PWS surface pressure observations with the 3DVar and 3DEnVar data assimilation schemes of AROME-France in one-month cycled experiments. A procedure has been presented in order to correct and select the observations dataset for purpose of assimilation.

The usability of surface pressure observations is related to the accuracy of their altitude metadata. When assimilated in a non-cycled AROME 3DVar experiment (the Monitor experiment), abnormal systematic differences in the OmB statistics were

shown in the pressure time series of 30 individual stations around Toulouse. The model topography is different from the true one, and the OmB statistics combine the error from the observation and the error from the model. The choice was made to correct the altitude of the PWS stations to the altitude they would have in the model. The station dependent bias has been found by averaging the OmBs from the Monitor experiment during one rolling month. The corrected observations have been used for both 3DVar and 3DEnVar experiments. The bias correction substantially reduces the range of individual geopotential bias

from ±1000 m$^2$ s$^{-2}$ to ±15 m$^2$ s$^{-2}$.

Then, unrealistic pressure observations were filtered. This was done by removing observations when their time series were too sparse; and when the standard deviation of the one-month geopotential departure between the PWS and the model was larger than 60 m$^2$ s$^{-2}$. This method ensures that observations belonging to time series with abnormal variations, due in particular to sudden changes in altitude or miscalibration, are removed.

The last pre-processing step was a thinning. With the use of the spatial Desroziers diagnostic, the set of observations was found to have limited spatial correlation of observation error. PWS observations were thinned by keeping one observation every





1.3 km in order to avoid (1) giving too much influence to PWSs observations when they are dense compared to conventional observing systems, which can make local minimisation of the cost function difficult and ultimately deteriorate the forecast (Eq. 1) and (2) propagating possible correlated observation errors in space, even if their existence has not been proven. After the preparation of the data, the bias correction, the quality control, and the thinning, there are an average of 27 578 pressure observations per hour ready to be assimilated, which represents 51 % of the initial observations.

In order to improve the pre-processing, there are several possibilities. Firstly, with regard to the bias correction, which can induce a slight bias of $-2.6\,\mathrm{m^2\,s^{-2}}$ on average. The bias correction uses the one-hour forecast of the Monitor experiment, which may be biased. Investigations over a longer period are needed to fully understand this bias. Another limitation of this method may lie in the consistency of assimilating into a 3DEnVar experiment PWS observations that have been bias-corrected thanks to OmB time series from the non-cycled 3DVar experiment. Other results may be expected using a bias correction which uses OmB time series from an non-cycled 3DEnVar experiment. In an operational context, a VarBC method could be a convenient and more consistent solution to routinely correct the bias of PWS observations.

Concerning the quality control (QC), the method used here sometimes suffers from not removing a wrong PWS surface pressure observation whose time series undergo a small but abnormal pressure variation (e.g. change of spatial position without any change in metadata). If such abnormal pressure variation persists over time, then the monthly rolling standard deviation of the innovation will keep growing hour by hour, and it will eventually exceed the rejection threshold. The QC will take some time to remove the wrong stations. More sophisticated QC could be done to remove erroneous observations faster and more accurately, with the help of the neighbours' variations. Indeed, a simple buddy check (as used by Vasiljevic et al., 2006) could help, while keeping in mind its difficulties to make the difference between erroneous variations and local physical patterns (as the one seen during the 14 September HPE). Finally, a more advanced technique, taking into account a temporal window to remove stations that are unrealistic during a few time steps (as MC20 method) could be beneficial. In the end, the supplementary QC presented here removed 1.6 % more observations, which is the same order of magnitude as the leave-one-out cross validation method presented by MC20. More investigation could be done on the utility of such method, moreover, if a superobbing method is tested in place of the thinning one. The necessity of the thinning procedure is not questioned in this study. However, the spatial Desroziers diagnostics has shown small observations correlation errors for the first 1 km distance bin. Experiments without thinning could be done in the future to understand how the assimilation scheme reacts when assimilating pressure observations from PWS in dense observations areas, like in the cities.

Making one-month cycled experiments has allowed us to see the overall impact of the assimilation of dense PWS surface pressure observations on analyses and forecasts of the AROME-France NWP system. When looking at statistics of geopotential OmB for SWS observations, the RMS OmB does not diminish when PWS observations are assimilated in the 3DVar DA scheme, due to systematic increases in the mean geopotential OmB for SWS observations. When removing the bias from the geopotential time series of individual SWSs (Appendix A), as it has been done previously for PWSs, the mean and the standard deviation of the OmB for SWSs are not degraded when assimilating PWS geopotential observations. This indicates that new methods to detect and reduce observational biases of SWS geopotential observations, bearing in mind that these anchoring observations are the link to reality, could allow to take advantage of PWS observations with AROME-France current 3DVar





DA scheme. On the contrary, when assimilating the new observations with the 3DEnVar DA scheme, the OmB RMS diminish up to −10.3 % for geopotential observations from SWSs situated in France.

To extend this study, the impact of assimilating PWS surface pressure observations on the quality of analyses could be studied with the Degrees of Freedom for Signal diagnostic (Chapnik et al., 2006). If the added value of assimilating surface pressure observation appears in local areas, further research should be conducted there.

Finally, the impact of the assimilation of a dense network of surface pressure observations related to a high precipitation event has been investigated. The event has been chosen due to its high intensity of 260 mm of rainfall in 3 h observed in the southern part of the Gard department and its associated local pressure variation. While the 3DVar has difficulties in reproducing the local pressure features, the 3DEnVar was found to create local scale surface pressure increments at the location where the PWS network observed the anomalies. The surface pressure increment is able to modify temperature, relative humidity and vertical velocity fields 50 min after the analysis. Then, the impact on rainfall has been studied by comparing forecasts to quantitative precipitation estimation products merging observations. The forecast launched from the 3DEnVar experiment assimilating PWS observations approximately 8 h before the most intense rainfall, improves both the location and the intensity of the rainfall maximum compared to a forecast launched from the 3DEnVar experiment without assimilating PWS observations. The maximum rainfall accumulation is increased by 22 % in the better forecast, which is closer to the observations. Still, this maximum rainfall accumulation remains 30 % lower from the observed one, underlining the need to further improve both analyses and the AROME model. Further research on other case studies is needed to confirm the positive results.

## Appendix A: Sensitivity of the Desroziers diagnostic to a geopotential bias correction

In a NWP system, because SWSs observations are, as said by Ingleby (2015), our link to reality, they are not bias-corrected. Thus, the mean geopotential OmA for SWSs ranges between 4.7 and 11.9 m$^2$ s$^{-2}$ for 3DVar and 3DVarP experiments, respectively. In this article, however, PWS observations are bias-corrected before the assimilation. As a consequence, the mean geopotential OmA for PWSs is −0.1 m$^2$ s$^{-2}$ in the 3DVarP experiment (Table 3) and the Desroziers diagnostic is lower for PWSs than for SWSs. To study the sensitivity of the bias correction on this diagnostic, an experiment in which SWS are bias-corrected was realised.

In this experiment called 3DVarP8corr (Table A1), SWS geopotential observations are assimilated after being corrected by subtracting the mean geopotential OmB computed on all time steps of one rolling month given by the Monitor experiment, similarly as for PWSs. The 3DVarP8corr experiment was run until the statistics, given in Table A2, were found to be stable over time, which took 8 days.

In 3DVarP8corr, the mean OmA for SWSs become close to zero (i.e. −0.2 m$^2$ s$^{-2}$) as wished. Even if the mean OmB for SWS are slightly diverging from the OmA statistics, the estimated Desroziers diagnostic significantly decreases from 25.0 m$^2$ s$^{-2}$ for 3DVarP8 to 10.7 m$^2$ s$^{-2}$ for 3DVarP8corr. The PWS bias correction explains why the Desroziers diagnostic is lower for PWSs than for SWSs in 3DVarP and 3DEnVarP experiments. This result confirms that the diagnostic cannot be used to compare the quality of two observing systems whose observations have not been corrected similarly.



**Table A1.** Overview of supplementary experiments run during 8 days between 6 September 2021 00:00 UTC and 14 September 2021 00:00 UTC.

| Experiments | Duration | Cycling | DA scheme | Use of PWS surface pressure observations | Use of SWS surface pressure observations |
|---|---|---|---|---|---|
| 3DVarP8 | 8 d | Yes | 3DVar | Assimilated after BC, QC and thinning | Assimilated without BC |
| 3DVarP8corr | | | | | Assimilated after BC |

**Table A2.** As Table 3 but for 3DVarP8 and 3DVarP8corr experiments. SWSs (resp. PWSs) provide 182 (resp. 27 997) observations per hour over 215 time steps.

| | 3DVarP8 | | | | 3DVarP8corr | | | |
|---|---|---|---|---|---|---|---|---|
| | SWS | | PWS | | SWS | | PWS | |
| | mean | std | mean | std | mean | std | mean | std |
| OmA | 11.4 | 21.6 | −0.1 | 13.0 | −0.2 | 8.8 | 0.0 | 12.9 |
| OmB | 10.4 | 40.2 | −0.6 | 34.6 | −1.9 | 35.1 | −0.8 | 34.8 |
| $\widetilde{\sigma_O}$ | 25.0 | | 13.9 | | 10.7 | | 13.9 | |

*Author contributions.* Alan Demortier: Formal investigations, methodology, analysis; resources; writing - original draft; writing - review and editing. Marc Mandement: Formal analysis; resources; writing - review and editing. Vivien Pourret: Formal analysis; software; validation; supervision; writing - review and editing. Olivier Caumont: validation; project administration; supervision; validation; writing - review and editing.

*Competing interests.* The authors declare that they have no conflict of interest.

*Code availability.* The code used for the assimilation experiments in AROME-France is owned by the members of the ACCORD consortium. This agreement allows each member of the consortium to license the shared ACCORD codes to academic institutions in their home countries for non-commercial research. Access to ACCORD codes, and codes used for the figures, can be obtained by contacting the corresponding author.

*Data availability.* World Meteorological Organization essential weather station data (SYNOP and SHIP reports), the radar mosaic at a 15 min time step, and AROME analyses and forecasts are available freely, online, in real time at https://donneespubliques.meteofrance.fr/ (last access: 15 June 2023); archives of RADOME data, radar mosaic data, ANTILOPE QPE, and AROME analyses and forecasts are available offline by making a request via this website. Personal weather station data are provided by and property of Netatmo and cannot be shared





according to their licence: https://dev.netatmo.com/legal (last access: 15 June 2023). The results of data assimilation experiments are available on request from the corresponding author solely for non-commercial research purposes and for reasonable data volumes.

*Acknowledgements.* This work was supported by the French National program Les Enveloppes Fluides et l'Environnement (LEFE), project ASMA.





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
