# Peer review of "Assimilation of surface pressure observations from personal weather stations in AROME-France"

_Natural Hazards and Earth System Sciences, 2023_

## Author Comment (AC1)

**Reply to referees - nhess-2023-103 - "Assimilation of surface pressure observations from personal weather stations in AROME-France"**

Parts added to the manuscript are in blue and deleted parts are in . Line numbers (L) in our replies are those of the preprint.

**Reply to anonymous referee #1**

**General comments**

> *The manuscript is generally well written. Like a number of other similar studies the sample size is rather limited, and on its own too short to prove that PWS data is beneficial.*

> We thank the first referee for the remarks and advice. The one-month period was inspired by other similar assimilation studies. Of course, a long-term experiment is usually conducted prior to operational implementation; such future study will be able to confirm the benefit of PWS pressure observation assimilation using the method described in the article.

> *As I understand it 3DVar is the operational system and 3DEnVar is a test system that may replace it at some point. In section 6 I would like to see some discussion of likely future work/changes. When might 3DEnVar become operational? When might use of Netatmo data become operational (or what further testing is required)?*

> The 3DEnVar is currently being tested at Météo-France in a real-time long-term experiment. If forecast quality scores are improved over the 3DVar system, it should be operational by June 2024. Netatmo data is currently freely downloaded by Météo-France using the Netatmo Weather API, but the data licence does not allow operational data assimilation at present. An ECMWF-EUMETNET pilot project aims to obtain Netatmo data over Europe that can be assimilated into NWP models. We have added some discussion at the end of section 6 (L513): For operational use, a few technical issues need to be discussed and distributed between meteorological centres, such as data flows, storage space, quality control and bias corrections, which would need to be encoded in the operational environment. The 3DEnVar would have to be operational for AROME-France. Only after these steps would the Netatmo data be assimilated in an AROME-France test chain controlled over several months.

> *Also what are the typical horizontal scales in the analysis of surface pressure of the two systems.*

> We have added the following sentence (L162) and a new figure in the Appendix: The propagation of the OmB from a surface pressure observation differ between the 3DVar and the 3DEnVar DA systems. If we consider that a typical horizontal scale in the surface pressure analysis is given by the distance at which the amplitude of a surface pressure analysis increment divided by its maximum value is below 1 %, then, this scale is approximately 200 km for the 3DVar scheme and 40 km for the 3DEnVar scheme (Fig. A). In 3DEnVar this typical horizontal scale can slightly vary depending on the meteorological situation. We have also completed the explanation of the localization in the section 2.2.

> *Using very high density data can cause convergence problems in a variational analysis system. Was there any sign of such problems (e.g., in number of iterations required and/or condition number)? What was the additional cost of assimilating the PWS data?*

> We have added the following sentence (L486): Dense observational network can be detrimental in the case of correlated observations errors and caused convergence problems and imbalanced fields (*Pourret et al.*, 2021; *Brousseau et al.*, 2016). Therefore, we applied a thinning at 1.3 km scale which reduce those effects.

For the convergence issues, we totally agree. The minimization in AROME-France using the 3DEnVar scheme has a limit of 50 iterations (*Michel and Brousseau*, 2021). In operational mode, the minimization always goes to the maximum number of iterations, which ensure near-identical computation time. To explain how the minimization works in research mode, the Eq. 1 has to be written $J(\boldsymbol{x}) = J^b(\boldsymbol{x}) + J^o(\boldsymbol{x})$, $J^b$ being the background term of the cost function. In research mode, the

[Figure]

**Figure A.** (a, b) Analysis increment of surface pressure resulting from the assimilation of a single observation of pressure at 1 m height in Blagnac, France and (c, d) amplitude of the longitudinal surface pressure analysis increment divided by its maximum value called normalised pressure increment (a, c) in an idealised 3DVar experiment and (b, d) in an idealised 3DEnVar experiment starting from the same guess. The black circle indicates a 25 km distance around the observation location.

40 minimization stops at the iteration number $i$ if the cost function background term gradient norm is less than $10^{-3}$ of the initial cost function background term gradient norm.Figure B shows the cost function background term gradient norm $\mathrm{DJ}(i) = \|J^b(x_i) - J^b(x_{i-1})\|$ as a function of the iteration number $i$ for the last minimization of the 1-month study period (on 5 October 2021 at 23:00 UTC) for the 3DEnVar and 3DEnVarP experiments. For this minimization in research mode, even if the cost function gradient norm is higher in 3DEnVarP w.r.t. 3DEnVar at the first iteration, the minimization takes only 1 more

45 iteration (Fig B). During the one-month study period, the average number of iterations increases by 0.6 in 3DEnVarP w.r.t. 3DEnVar. We have added (L265): When pressure observations from PWSs were added to the AROME-France assimilation cycles, no convergence problems were reported.

  *Personally I would have preferred to see the statistics in pressure units (rather than geopotential units) throughout, but I can see the logic of using the assimilated variable.*

50 > We understand, as we are also more familiar with pressure units, and you're right, we prefer to use the geopotential units since it is the assimilated variable.

[Figure]

**Figure B.** Evolution of the cost function background term gradient norm DJ as a function of the iteration number during the minimization of the 3DEnVarP and 3DEnVar experiments on 5 October 2021 at 23:00 UTC.

*I have made some suggestions to improve the English (but it is better than my French).*

> We thank the reviewer for his constructive corrections, which we have taken into account in the revised article.

**Detailed comments**

55     *line 1 [PWSs] 'have proved their ability' This seems too definite to me, if fully proved why do we need another paper on the subject? Reword.*

> We have reworded the first sentence:  are able to describe pressure patterns at the surface, such as those associated with convective events, in more detail than with standard weather stations (SWSs) only.

60     *Section 1*
    *I suggest adding at least two of the four references below (two of which use Netatmo data): Nipen et al. (2020); Sgoff et al. (2022); Bell et al. (2013); Hahn et al. (2022).*

> We have added two of them in the sentence on L20: Crowdsourced observations are much more numerous than observations of standard weather stations (SWSs) currently assimilated in NWP models, while having a more heterogeneous quality and
65     less metadata *(Nipen et al.*, 2020; *Sgoff et al.*, 2022).

    *24 'combined to SWS' - 'combined with SWS'*

> We have corrected it.

    *26-27 'MC20 showed that PWS surface pressure observations allow multiplying by an order of magnitude of 100 ... current state.' - 'MC20 showed that use of PWS allows approximately 100 times more surface pressure*
70     *observations to be assimilated.' (Better English.)*

> We have corrected it.

    *As noted in the general comments in practice it is possible to have 'too many' observations and this should be addressed somewhere in the manuscript.*

> Our answer is in the general comments section.

75    *28 'These observations, when simply spatialized' - unclear, 'spatialized' not a word? Possibly 'when regularly spaced'?*

> The sentence has been clarified:  These observations, when used to produce gridded analyses of mean sea level pressure (MSLP), are the
80    observations that most reduce (of the three observed variables, during four stormy days) the root-mean-square error of these analyses compared to analyses using only SWSs.

   *36 'World Meteorological Organization (WMO) Regional Basic Observing Network (WMO, 2021)' Instead of RBON I would refer to GBON (Global Basic Observing Network) https://community.wmo.int/en/activity-areas/wigos/gbon - a relatively recent concept.*

85    > A reference to the Global Basic Observing Network (GBON) and its link with the RBONs has been added (L37): from the World Meteorological Organization (WMO) Regional Basic Observing Networks of all WMO Regions, of which the Global Basic Observing Network is a subset(WMO, 2021).
The reference to the RBONs is kept given the fact that globally, the GBON is a subset of the RBONs according to the link "GBON Concept" in the webpage indicated by the referee (WMO, 2021): "The GBON is a subset of the surface-based subsys-
90    tem of WIGOS [...] to contribute to meeting the requirements of Global NWP [...]. The GBON is the foundation upon which the Regional Basic Observing Networks (RBONs) are built to respond to requirements of a broader range of WMO application areas, including further requirements of Global NWP beyond the essential base provided by the GBON. Hence all GBON stations/platforms and their observing programmes (variables and schedules) are included in the respective RBON of the Region in which they are operating.".

95    *82 'Sect. 3' perhaps personal preference, I would write Section in full.*

> The abbreviation has been maintained according to the publisher's guidelines (*Copernicus Publications*, 2023).

   *87 'represents 717 cycled hours of simulation for each experiment (without two time steps' - 'represents 717 hours for each experiment (excluding two time steps' suggestion*

> We have corrected it.

100    *96 'on ships (sending SHIP reports)' - 'on ships and buoys'?*

> We have not modified it, because moored and drifting buoys are not included in SWSs.

   *Figure 1 caption: 'emitting' - 'providing'*
   *'metropolitan France' - in English this means Paris or 'urban France', not the intention. Just 'France'.*

> We have corrected it.

105    *137 'J is incremental' - is it calculated at full forecast model resolution? Sometimes the increments are at lower resolution. Please clarify in the main text.*

> We have added on L137: , at full model resolution. If $\mathbf{B}$ is computed at a lower resolution than the model, interpolation operators from low to high resolution are applied.

   *160 'reduces the propagation of the innovation at a shorter spatial distance' - 'reduces the propagation of the innovation at larger distances' or 'limits the propagation of the innovation to shorter distances'.*

110

> We have added: limits the propagation of the OmB to shorter distances. The word "innovation" is replaced by OmB due to a comment of the second referee.

*200 'Andersson et al., 1991' I would remove this reference, old OSEs are not really useful now.*

> We have removed it.

*201 'Making OSEs allows us to' - 'OSEs allow us to'*

> We have corrected it.

*212 'interpolated at round hours, which concomitantly removes' 'interpolated to round hours, which also removes'*
*I don't understand the bit about removing sparse observations.*

> The sentence has been modified:  interpolated to round hours, using the method described in Sect. 2.1.2 which removes some observations.

*217 'concomitantly' - delete*

> We have deleted it.

*224 'the Figure 3' - delete 'the'*

> We have deleted it.

*227ff 'To implement a geopotential bias correction for PWS data based on AROME ... changes the altitude of the station in the assimilation scheme.' One of the main causes of surface pressure biases is errors in the station altitude.*
*One cause of error in station altitude is use of uncorrected GPS heights: "GNSS systems use a reference ellipsoid, and so the 'undulation of the geoid' must be taken into account to get heights relative to sea level. (Appendix 2 describes a nearly 40 m error at a radiosonde site where this was not taken into account.)" from (Pauley and Ingleby, 2022).*

> We have added (L118): One of the main sources of surface pressure bias is errors in the station altitude, caused in part by uncorrected GPS heights (*Pauley and Ingleby*, 2022).

*248 '1 is randomly drawn' - 'one is randomly drawn'*
*248 'generally never' - 'not usually'*
*289 'perturbed' - 'unsettled'?*
*312 'meridian' - 'meridional'*
*Figure 6 caption 'improvement' - presumably with respect to the control?*

> We have corrected it.

*Figure 7. The horizontal (average) lines partially obscure the score values. They should either be removed entirely or restricted to the side of the figure.*
*Figure 7 caption. Add that higher values are better.*

> We have updated Fig. 7 with new bars at the side of the figure, and we have added that higher values are better.

*359 'developed' - 'shown'*

145  *264 'dumping' - 'giving'*

*387 'which' - 'this'*

*Figure 8 caption 'black contours' - 'black edges' or 'black outlines'*

*300 'Once known ...' awkward. Perhaps 'Given the pressure increments the influence on other variables is explored.'*

150  *410 'Spatial size ...' - 'The surface pressure differences in Fig. 10e have smaller scale than the analysis increments in Fig. 10a.'*

*'5.4 Impact on forecasted rainfall' - remove 'ed' from forecasted*

> We have corrected it.

*473/4 'a VarBC method could be ... correct the bias of PWS observations.' There are some caveats to this. Ideally*
155  *there should be a good proportion of (uncorrected) anchor observations, but SWS are swamped by PWS data. How much this matters in practice for surface pressure isn't clear. Eyre, J.R. (2016), Observation bias correction schemes in data assimilation systems: a theoretical study of some of their properties. Q.J.R. Meteorol. Soc., 142: 2284-2291. https://doi.org/10.1002/qj.2819*

> We agree with the reviewer. This question was not tackled in the article, and should be treated when the VarBC would be
160  tested. We have added: while being aware of the need to maintain sufficient anchor observations from SWSs (*Eyre*, 2016).

**Reply to anonymous referee #2**

*I think this is a very well-written, clear description of a very important set of experiments. The number of non-professionally sited surface instruments is exploding. One would hope that the number of obs could counterbalance their less than optimal siting or poor measurement error characteristics. One of the easiest obs to do this with is*
165  *surface pressure since it is much less severely impacted by poor siting. This study describes a carefully designed set of experiments. Details of bias correction, and thinning are important and are presented clearly. A number of summary verification statistics for both analyses, and forecasts of varying length are included. The diagnostics of the heavy rain event are good choices and provide insight into how the personal surface pressure obs may have improved things. I have no major concerns about the manuscript, but I will append a short list of minor points.*
170  *Thanks to the authors for presenting such a nice study on a subject I have been interested in for decades.*

> We would like to thank the second anonymous reviewer for his review and comments.

*1. Line 94: replace 'gather' with 'include'*

> It has been modified.

*2. Sentence starting at line 99: It was unclear to me how these numbers differed from those in the previous sentence.*
175  *Please clarify.*

> We have clarified the difference between AROME-France domain and the France subdomain. As France is a fraction of the AROME-France domain, only a part of these stations are located in France.

*3. Sentence starting at line 126: I wasn't sure what the SURFEX model was here and what the relation of the Meso-NH was to other models discussed.*

180  > We agree that this was not clear. We have modified the sentence:  Physical parametrizations come from the Meso-NH research model (*Lac et al.*, 2018). At the surface, AROME-France is coupled with the SURFEX model (*Masson et al.*, 2013).

*4. Line 128: 'thereafter' to 'hereafter'*

> It has been modified.

185     *5. Line 135: I don't think 'determines' is quite the right word. Maybe 'estimates' or 'approximates'?*

> It has been modified.  approximates

*6. Line 143: 'multiple observations during the assimilation window'. I was confused here. I thought you were averaging the two nearest obs around each hour so that there could not be multiple obs during an assimilation window of 1-hour?*

190 > You are right, PWS multiple observations during the assimilation window are dealt with the method indicated on L110–L112. We have corrected the sentence to talk about SWS only: for stationary stations ( SWS except ships) providing multiple observations during the assimilation window. Only PWS observations valid at $t_0$ are used, following the method described in Sect. 2.1.2.

*7. Line 148: This sudden mention of geopotential obs was confusing. I suggest having the discussion of the use of*
195 *geopotential versus pressure before you introduce the associated error constant. I will admit that I was never quite clear on the geopotential vs surface pressure uses either. I suspect that the paper describes it correctly, but maybe not quite as clearly in one place as it could.*

> As suggested, the line at which the standard deviation of the geopotential error is mentioned has been moved after the discussion on the geopotential. As mentioned by the first reviewer, the use of both surface pressure and geopotential may be
200 confusing. However, as we mentioned, the use of the geopotential is essential as it is the variable used to assimilate the surface pressure observations in the AROME-France model. We hope that the formulation used is as clear as possible.

*8. Sentence starting at line 160: The discussion of the localization needs more clarity, and I suggest including an equation to make it clear. Is the localization something like Gaspari-Cohn with a halfwidth that varies linearly with height? Or is the localization itself linear?*

205 > We agree with the reviewer that the discussion on the localization was rather limited. The localized matrix $\hat{\mathbf{B}}_e$ is obtained making an element-to-element product (known as a Hadamard–Schur product) between $\mathbf{B}_e$ and a correlation matrix $\mathbf{L}$ of the same dimension (Eq. 1).

$$\hat{\mathbf{B}}_e = \mathbf{B}_e \circ \mathbf{L}, \tag{1}$$

The correlation functions in the $\mathbf{L}$ matrix are quasi-Gaussian with horizontal and vertical localizations (*Montmerle et al.*, 2018).
210 The specified length of localization corresponds to the half width at half maximum of the quasi-Gaussian function.

*9. Sentence starting on line 163: This seems out of place in the middle of the geopotential discussion and might be better placed somewhere else.*
*10. Sentence starting at line 167: I struggle to understand exactly what this sentence implies. If possible, try to make this distinction a bit more clear.*

215 > We have created a new subsection dedicated to the geopotential discussion (after introducing the SWSs and the PWSs). We have tried to rephrase as much as possible.  [...] to refer to the same observations. The surface pressure will be used when studying the observations, whereas the corresponding geopotential at the level of the observed
220 station pressure will be the variable used when looking at assimilation statistics.

*11. Line 174: 'reduce this correlation' to 'reduce the negative impact of this correlation'. In fact, your later results suggest that there is almost no correlation and that the real benefit of thinning is to reduce the dominance of the PWS surface pressure observations.*

> We have included the proposed modification. In this section we present the hypothesis made for the R matrix, being diagonal, and why it is common to use a thinning method. However, we agree that the thinning here was mainly done to reduce the dominance of the PWS observations.

*12. Line 175: You introduce the term 'innovation' but then confuse the definition with the mention of OmA and OmB in the next sentence. I don't think you ever use the term innovation again, so maybe you don't need to define it?*

> Indeed, we have removed the term "innovation" which was redundant with the term OmB. Furthermore, we have modified this sentence: The difference between an observation and its corresponding value in the background, called observation minus background (OmB), informs us of the concordance between both the observations and the model 1 h forecast.

*13. Line 201 and subsequent places: I have a long history of criticizing the use of analysis error statistics because of the potential problem of overfitting. I don't think that is a huge problem here, but I caution the authors to be very careful when looking at analysis errors, especially when there is a large number of a particular kind of observation.*

> We thank the reviewer for the wise advice. We agree that we may be very careful when looking at analysis errors, and we have mainly discussed OmBs to avoid the problem of correlations between observations and model.

*14. Line 204: The description of the 'monitoring' experiment was not as clear as it could be. If I understand correctly, it does not cycle because it uses the background from one of the cycling cases. You say here that the raw pressures are assimilated, but is that really the purpose of the monitor? Isn't it only used to look at OmB? My confusion might suggest a way to clarify the description.*

> Exactly, it does not cycle because it uses the background from the operational cycling experiment. The term *assimilating* was confusing as we were only doing a monitoring. We have corrected it:  , that has been monitoring We have also added a sentence to clarify the use of the monitoring experiment. This monitoring experiment will be used to compute both the bias correction and the quality control of the PWS observations.

*15. Line 249-50: You are correct that these are independent instruments. I guess the only reason to expect correlated error is instrument systematic bias (all instruments always too high) or correlation of the altitude error of the model grid at the subgrid scale?*

> We thank the reviewer for this interesting comment. We have added it to the section: For surface pressure, correlated errors could come from instrument systematic bias or correlation of the altitude error of the model grid at the subgrid scale.

*16. Line 287: Phrase starting with "which" was unclear to me.*

> We have rephrased the sentence:  , while the 3DEnVarP is not systematically closer to 0 than the 3DEnVar for the geopotential mean OmB.

*17. Line 376: First phrase starting with "As" was unclear to me.*

> We have modified the sentence, to make it clearer:  Since the four experiments start from a different background, this comparison is intended to illustrate the spatial structure of surface pressure increments and, for each observation, the associated change in analysis departure with respect to background departure.

*18. Figure 8: There is a lot of information here and I had trouble reading it. It might be better to separate into two panels, one with the obs and the other with the radar and the wind info. However, even the current panel mostly supports your points.*

> We have removed the information about the wind (direction and intensity) in the updated figure. We think it is useful to keep the radar reflectivities and the mean sea level pressure (MSLP) observations in the same figure, to easily compare structures of observed MSLP field with observed reflectivity field.

*19. Line 439: Any ideas on why there is a degradation for the small threshold?*

> The score is calculated on a single forecast. To interpret this degradation correctly, the same score would have to be calculated for another heavy precipitation event, or over a different time period.

**References**

270    Bell, S., D. Cornford, and L. Bastin, The state of automated amateur weather observations, *Weather*, *68*, 36–41, 2013.

Brousseau, P., Y. Seity, D. Ricard, and J. Léger, Improvement of the forecast of convective activity from the AROME-France system, *Quarterly Journal of the Royal Meteorological Society*, *142*(699), 2231–2243, https://doi.org/10.1002/qj.2822, 2016.

Copernicus Publications, Natural Hazards and Earth System Sciences–Submissions, https://www.natural-hazards-and-earth-system-sciences.net/submission.html (last access: 7 November 2023), 2023.

275    Eyre, J. R., Observation bias correction schemes in data assimilation systems: a theoretical study of some of their properties, *Quarterly Journal of the Royal Meteorological Society*, *142*(699), 2284–2291, https://doi.org/10.1002/qj.2819, 2016.

Hahn, C., et al., Observations from personal weather stations—eumetnet interests and experience, *Climate*, *10*(12), https://doi.org/10.3390/cli10120192, 2022.

Lac, C., et al., Overview of the Meso-NH model version 5.4 and its applications, *Geoscientific Model Development*, *11*(5), 1929–1969,
280    https://doi.org/10.5194/gmd-11-1929-2018, 2018.

Masson, V., et al., The SURFEXv7.2 land and ocean surface platform for coupled or offline simulation of earth surface variables and fluxes, *Geoscientific Model Development*, *6*(4), 929–960, https://doi.org/10.5194/gmd-6-929-2013, 2013.

Michel, Y., and P. Brousseau, A square-root, dual-resolution 3DEnVar for the AROME model: Formulation and evaluation on a summertime convective period, *Monthly Weather Review*, *149*(9), 3135–3153, https://doi.org/10.1175/MWR-D-21-0026.1, 2021.

285    Montmerle, T., Y. Michel, E. Arbogast, B. Ménétrier, and P. Brousseau, A 3D ensemble variational data assimilation scheme for the limited-area AROME model: Formulation and preliminary results, *Quarterly Journal of the Royal Meteorological Society*, *144*(716), 2196–2215, https://doi.org/10.1002/qj.3334, 2018.

Nipen, T. N., I. A. Seierstad, C. Lussana, J. Kristiansen, and H. Ø, Adopting citizen observations in operational weather prediction, *Bulletin of the American Meteorological Society*, *101*(1), E43–E57, https://doi.org/10.1175/BAMS-D-18-0237.1, 2020.

290    Pauley, P. M., and B. Ingleby, *Assimilation of In-Situ Observations*, pp. 293–371, Springer International Publishing, Cham, https://doi.org/10.1007/978-3-030-77722-7_12, 2022.

Pourret, V., J.-F. Mahfouf, V. Guidard, P. Moll, A. Doerenbecher, and B. Piguet, Variational bias correction for Mode-S aircraft derived winds, *Tellus A: Dynamic Meteorology and Oceanography*, *73*(1), 1886,808, https://doi.org/10.1080/16000870.2021.1886808, 2021.

Sgoff, C., W. Acevedo, Z. Paschalidi, S. Ulbrich, E. Bauernschubert, T. Kratzsch, and R. Potthast, Assimilation of crowd-sourced surface
295    observations over Germany in a regional weather prediction system, *Quarterly Journal of the Royal Meteorological Society*, *148*(745), 1752–1767, https://doi.org/10.1002/qj.4276, 2022.

World Meteorological Organization, Global Basic Observing Network concept, https://wmoomm.sharepoint.com/:b:/s/wmocpdb/EfMIe61iiPdDgCTm6aFFpnUB46D6SIImYKvjXCMegZHCCQ?e=T3Vtgq (last access: 7 November 2023), 2021.

---

## Author Response (AR2)

**Second reply to referees - nhess-2023-103 - "Assimilation of surface pressure observations from personal weather stations in AROME-France"**

Parts added to the manuscript are in blue and deleted parts are in . Line numbers (L) in our replies are those of the version after the first revision.

**Reply to anonymous referee #1**

*General*

*I would like to thank the authors for adding Appendix A. The difference in length scales - 40 km vs 200 km is dramatic - and I feel this should be mentioned in the summary. My detailed comments below are mainly suggestions for improving the wording, with some requests for clarification.*

> We thank the referee for the remarks and advices. Indeed, the difference in length scales between 3DVar and 3DEnVar DA scheme is huge. We have mentioned it in the summary (L530): This reduction of length scale from 200 km in 3DVar to 40 km in 3DEnVar for a single observation is illustrated in Appendix A

*line 9: 'reduces by -10.3 % the root-mean-square departure' remove minus sign before 10.3 %.*

> We have corrected it.

*10: 'improvement is observed up to 9 h of forecast' - '... observed over the first 9 h of the forecasts' (or perhaps 'the first 6-9 h' for eg)*

> We have corrected it.

*17: 'accurately modelling' - 'accurately estimating'*

> We have corrected it.

*35/36 The RBON/GBON sentence now seems awkward to me. Could say 'Regional/Global Basic Observing Networks exchanged via the WMO Global Telecommunication System'*

> We have corrected it.

*144 'informs us of the concordance between both the ...' 'concordance' is an unusual word in English, perhaps 'provides a comparison between the ...'*

> We have corrected it.

*152 'screening' is there any check on the height difference between the station and the model? (It can be large in mountainous areas.)*

> To answer this good question, the following sentence has been added L156: Two other screening steps that are important for mass observations (surface pressure and geopotential height) include checks for missing station altitude, and compliance with WMO observation standards: if the reported observation level is far from the true station level, $\sigma_o$ is inflated (more details in Sect. 4.4.1 of *ECMWF*, 2023). Height differences between stations and the ground level of the model, which can be large in mountainous areas, are not managed by the screening but rather by the observation operator described in Sect. 2.2.1.

*192-194 'The allocated part of the R matrix for PWSs ... the prescribed observation errors.' Perhaps just 'the same' (instead of 'the prescribed observation errors') if I understood correctly. It might be better to give PWSs larger errors?*

> We have replaced "the prescribed observation errors" by "the same". The largest estimated $\sigma_o$ for PWSs given by Desroziers diagnostic equals $19.7\,\text{m}^2\,\text{s}^{-2}$ in 3DEnVarP experiment (Table 3) which is much lower than the prescribed one which equals $8g = 78.5\,\text{m}^2\,\text{s}^{-2}$. It suggests that the prescribed $\sigma_o$ is maybe still overestimated. However, it is true that it might be better to give PWSs larger $\sigma_o$ than SWSs: this has not been tested. A sentence has been added L195: The effect of different $\sigma_o$ between PWS and SWS observations could be investigated in a sensitivity study.

*209,210 'the spatial correlation of observations covariance error' - 'the spatial correlation of observation error'*

> We have corrected it.

*237 'less than 150 m from higher to lower point' - 'less than 150 m from lowest to highest point'*

> We have corrected it.

*315,316 'The large increase in OmA geopotential mean when PWS observations are assimilated could be explained by the PWS bias correction method, which uses OmB and not OmA statistics for it.' I suspect that using OmA for the bias estimate wouldn't help much - it is more about the relative numbers (and weight) of the two subsets (Eyre, 2016).*

We have modified L315–316: The large increase in OmA geopotential mean when PWS observations are assimilated could be explained by the PWS bias correction method, which uses OmB  statistics, and thus contains model or other observation biases (as SWS ones). As 129 times more PWS observations than SWS observations are assimilated, with the same prescribed observation error, the 3DVar long range PWS increments overwhelm the SWS ones, which questions the relative number of observations in each subset and its impact in different analysis methods.

*327 'some limits' - 'some limitations'*

> We have corrected it.

*345 'partly compensates for the significant degradation associated with the transition from a 3DEnVar system to the 3DVar.' should this be 'to a 3DEnVar system from the 3DVar'?*
*Is part of the issue that the 3DVar fields are smoother than the 3DEnVar fields? (Linked to discussion about finer localisation at line 295?).*
*It is hard work keeping track of the discussion (not helped by the names just differing by 'En') if it is possible to simplify parts of section 4 while still making the main points that would be good.*

> We have corrected the inversion between "from" and "to". In fact, the degradation is due to the change of the B matrix from the 3DVar to the 3DEnVar DA and could be due to the finer localization carried out within the 3DEnVar DA scheme, without changing the prescribed observation errors (L295).
We have simplified this section by removing:

– a sentence regarding OmA of 3DVar and 3DVarP (L299):

– redundant explanation (L303):

– a sentence regarding OmA of 3DEnVar and 3DEnVarP:  .

75    – a few repetitions to lighten the style.

*530 'the 3DEnVar was found to create local scale surface pressure increments ...' mention Appendix A and length scales*

> We have added (L530): [...] the 3DEnVar was found to create reduced length  scale surface pressure increments at the location where the PWS network observed the anomalies. This reduction of length scale from 200 km in 3DVar to 40 km in
80   3DEnVar for a single observation is illustrated in Appendix A .

**References**

ECMWF, *IFS Documentation CY48R1 - Part I: Observations*, chap. 1, pp. 1–88, 1, ECMWF, https://doi.org/10.21957/0f360ba4ca, 2023.